# INFORMATION FLOW IN SELF-SUPERVISED LEARNING

## ABSTRACT

In this paper, we provide a comprehensive toolbox for understanding and enhancing self-supervised learning (SSL) methods through the lens of matrix information theory. Specifically, by leveraging the principles of matrix mutual information and joint entropy, we offer a unified analysis for both contrastive and feature decorrelation based methods. Furthermore, we propose the matrix variational masked auto-encoder (M-MAE) method, grounded in matrix information theory, as an enhancement to masked image modeling. The empirical evaluations underscore the effectiveness of M-MAE compared with the state-of-the-art methods, including a $3.9\%$ improvement in linear probing ViT-Base, and a $1\%$ improvement in fine-tuning ViT-Large, both on ImageNet.

## 1 INTRODUCTION

Self-supervised learning (SSL) has demonstrated remarkable advancements across various tasks, including image classification and segmentation, often surpassing the performance of supervised learning approaches (Chen et al., 2020; Caron et al., 2021; Li et al., 2021; Zbontar et al., 2021; Bardes et al., 2021). Broadly, SSL methods can be categorized into three types: contrastive learning, feature decorrelation based learning, and masked image modeling.

One prominent approach in contrastive self-supervised learning is SimCLR (Chen et al., 2020), which employs the InfoNCE loss (Oord et al., 2018) to facilitate the learning process. Interestingly, Oord et al. (2018) show that InfoNCE loss can serve as a surrogate loss for the mutual information between two augmented views. Unlike contrastive learning which needs to include large amounts of negative samples to "contrast", another line of work usually operates without explicitly contrasting with negative samples which we call feature decorrelation based learning. Recently, there has been a growing interest in developing feature decorrelation based SSL methods, e.g., BYOL (Grill et al., 2020), SimSiam (Chen & He, 2021), Barlow Twins (Zbontar et al., 2021), VICReg (Bardes et al., 2021), etc. These methods have garnered attention from researchers seeking to explore alternative avenues for SSL beyond contrastive approaches.

On a different note, the masked autoencoder (MAE) (He et al., 2022) introduces a different way to tackle self-supervised learning. Unlike contrastive and feature decorrelation based methods that learn useful representations by exploiting the invariance between augmented views, MAE employs a masking strategy to have the model deduce the masked patches from visible patches. Therefore, the representation of MAE carries valuable information for downstream tasks.

At first glance, these three types of self-supervised learning methods may seem distinct, but researchers have made progress in understanding their connections. Garrido et al. (2022) establish a duality between contrastive and feature decorrelation based methods, shedding light on their fundamental connections and complementarity. Additionally, Balestriero & LeCun (2022) unveil the links between popular feature decorrelation based SSL methods and dimension reduction methods commonly employed in traditional unsupervised learning. These findings contribute to our understanding of the theoretical underpinnings and potential applications of feature decorrelation based SSL techniques. However, compared to connections between contrastive and feature decorrelation based methods, the relationship between MAE and contrastive or feature decorrelation based methods remains largely unknown. To the best of our knowledge, Zhang et al. (2022b) is the only paper that relates MAE to the alignment term in contrastive learning.

Though progress has been made in understanding the existing self-supervised learning methods, the tools used in the literature are diverse. As contrastive and feature decorrelation based learning

usually use two augmented views of the same image, one prominent approach is analyzing the mutual information between two views (Oord et al., 2018; Shwartz-Ziv et al., 2023; Shwartz-Ziv & LeCun, 2023). A unified toolbox to understand and improve self-supervised methods is needed. Recently, Bach (2022); Skean et al. (2023) have considered generalizing the traditional information-theoretic quantities to the matrix regime. Interestingly, we find these quantities can be powerful tools in understanding and improving existing self-supervised methods regardless of whether they are contrastive, feature decorrelation-based, or masking-based (He et al., 2022).

Taking the matrix information theoretic perspective, we analyze some prominent contrastive and feature decorrelation based losses and prove that both Barlow Twins and spectral contrastive learning (HaoChen et al., 2021) are maximizing mutual information and joint entropy. These claims are crucial for analyzing contrastive and feature decorrelation based methods, offering a cohesive and elegant understanding. More interestingly, the same analytical framework extends to MAE as well, wherein the concepts of mutual information and joint entropy gracefully degenerate to entropy. Propelled by this observation, we augment the MAE loss with matrix entropy, giving rise to our new method, Matrix variational Masked Auto-Encoder (M-MAE). Empirically, M-MAE stands out with commendable performance. Specifically, it has achieved a $3.9\%$ improvement in linear probing ViT-Base, and a $1\%$ improvement in fine-tuning ViT-Large, both on ImageNet. This empirical result not only underscores the efficacy of M-MAE but also accentuates the potential of matrix information theory in ushering advancements in self-supervised learning paradigms.

In summary, our contributions can be listed as follows:

- We use the matrix information-theoretic tools to understand existing contrastive and feature decorrelation based self-supervised methods.
- We introduce a novel method, M-MAE, which is rooted in matrix information theory, enhancing the capabilities of standard masked image modeling.
- Our proposed M-MAE has demonstrated remarkable empirical performance, showcasing a notable improvement in self-supervised learning benchmarks.

## 2 BACKGROUND

### 2.1 MATRIX INFORMATION-THEORETIC QUANTITIES

In this section, we shall briefly summarize the matrix information-theoretic quantities that we shall use in this paper. We shall first provide the definition of (matrix) entropy as follows:

**Definition 2.1** (Matrix-based $\alpha$-order (Rényi) entropy (Skean et al., 2023)). Suppose matrix $\mathbf{K}_1 \in \mathbb{R}^{n \times n}$ which $\mathbf{K}_1(i, i) = 1$ for every $i = 1, \cdots, n$. $\alpha$ is a positive real number. The $\alpha$-order (Rényi) entropy for matrix $\mathbf{K}_1$ is defined as follows:

$$\mathrm{H}_\alpha(\mathbf{K}_1) = \frac{1}{1-\alpha} \log \left[ \mathrm{tr} \left( \left( \frac{1}{n} \mathbf{K}_1 \right)^\alpha \right) \right],$$

where $\mathbf{K}_1^\alpha$ is the matrix power.

The case of $\alpha = 1$ recovers the von Neumann (matrix) entropy, i.e.

$$\mathrm{H}_1(\mathbf{K}_1) = -\mathrm{tr} \left( \frac{1}{n} \mathbf{K}_1 \log \frac{1}{n} \mathbf{K}_1 \right).$$

Using the definition of matrix entropy, we can define matrix mutual information and joint entropy as follows.

**Definition 2.2** (Matrix-based mutual information). Suppose matrix $\mathbf{K}_1, \mathbf{K}_2 \in \mathbb{R}^{n \times n}$ which $\mathbf{K}_1(i, i) = \mathbf{K}_2(i, i) = 1$ for every $i = 1, \cdots, n$. $\alpha$ is a positive real number. The $\alpha$-order (Rényi) mutual information for matrix $\mathbf{K}_1$ and $\mathbf{K}_2$ is defined as follows:

$$\mathrm{I}_\alpha(\mathbf{K}_1; \mathbf{K}_2) = \mathrm{H}_\alpha(\mathbf{K}_1) + \mathrm{H}_\alpha(\mathbf{K}_2) - \mathrm{H}_\alpha(\mathbf{K}_1 \odot \mathbf{K}_2).$$

**Definition 2.3** (Matrix-based joint entropy). Suppose matrix $\mathbf{K}_1, \mathbf{K}_2 \in \mathbb{R}^{n \times n}$ which $\mathbf{K}_1(i, i) = \mathbf{K}_2(i, i) = 1$ for every $i = 1, \cdots, n$. $\alpha$ is a positive real number. The $\alpha$-order (Rényi) joint-entropy for matrix $\mathbf{K}_1$ and $\mathbf{K}_2$ is defined as follows:

$$\mathrm{H}_\alpha(\mathbf{K}_1, \mathbf{K}_2) = \mathbf{H}_\alpha(\mathbf{K}_1 \odot \mathbf{K}_2),$$

where $\odot$ is the (matrix) Hadamard product.

Another important quantity is the matrix KL divergence defined as follows.

**Definition 2.4** (Matrix KL divergence (Bach, 2022)). *Suppose matrix* $\mathbf{K}_1, \mathbf{K}_2 \in \mathbb{R}^{n \times n}$ *which* $\mathbf{K}_1(i,i) = \mathbf{K}_2(i,i) = 1$ *for every* $i = 1, \cdots, n$. *The Kullback-Leibler (KL) divergence between these two matrices* $\mathbf{K}_1$ *and* $\mathbf{K}_2$ *is defined as*

$$\mathrm{KL}\left(\mathbf{K}_1 \parallel \mathbf{K}_2\right) = \mathrm{tr}\left[\mathbf{K}_1 \left(\log \mathbf{K}_1 - \log \mathbf{K}_2\right)\right].$$

## 2.2 CANONICAL SELF-SUPERVISED LEARNING LOSSES

We shall recap some canonical losses used in self-supervised learning. As we roughly characterize self-supervised learning into contrastive learning, feature decorrelation based learning, and masked image modeling. We shall introduce the canonical losses used in these areas sequentially.

In contrastive and feature decorrelation based learning, people usually adopt the Siamese architecture, namely using two parameterized networks: the online network $f_\theta$ and the target network $f_\phi$. To create different perspectives of a batch of $B$ data points $\{\mathbf{x}_i\}_{i=1}^{B}$, we randomly select an augmentation $\mathcal{T}$ from a predefined set $\tau$ and use it to transform each data point, resulting in new representations $\mathbf{z}_i^{(1)} = f_\theta(\mathcal{T}(\mathbf{x}_i)) \in \mathbb{R}^d$ and $\mathbf{z}_i^{(2)} = f_\phi(\mathbf{x}_i) \in \mathbb{R}^d$ generated by the online and target networks, respectively. We then combine these representations into matrices $\mathbf{Z}_1 = [\mathbf{z}_1^{(1)}, \ldots, \mathbf{z}_B^{(1)}]$ and $\mathbf{Z}_2 = [\mathbf{z}_1^{(2)}, \ldots, \mathbf{z}_B^{(2)}]$. In masked image modeling, people usually adopt only one branch and do not use Siamese architecture.

The idea of contrastive learning is to make the representation of similar objects align and dissimilar objects apart. One of the widely adopted losses in contrastive learning is InfoNCE loss (Chen et al., 2020), which is defined as follows:

$$\mathcal{L}_{\mathrm{InfoNCE}} = -\frac{1}{2}\left(\sum_{i=1}^{B} \log \frac{\exp\left((\mathbf{z}_i^{(1)})^\top \mathbf{z}_i^{(2)}\right)}{\sum_{j=1}^{B} \exp\left((\mathbf{z}_i^{(1)})^\top \mathbf{z}_j^{(2)}\right)} + \sum_{i=1}^{B} \log \frac{\exp\left((\mathbf{z}_i^{(2)})^\top \mathbf{z}_i^{(1)}\right)}{\sum_{j=1}^{B} \exp\left((\mathbf{z}_i^{(2)})^\top \mathbf{z}_j^{(1)}\right)}\right). \quad (1)$$

As the InfoNCE loss may be difficult to analyze theoretically, HaoChen et al. (2021) then propose spectral contrastive loss as a good surrogate for InfoNCE. The loss is defined as follows:

$$\sum_{i=1}^{B} \parallel \mathbf{z}_i^{(1)} - \mathbf{z}_i^{(2)} \parallel_2^2 + \lambda \sum_{i \neq j} ((\mathbf{z}_i^{(1)})^\top \mathbf{z}_j^{(2)})^2, \quad (2)$$

where $\lambda$ is a hyperparameter.

The idea of feature decorrelation based learning is to learn useful representation by decorrelating features and do not explicit distinguishes negative samples. Some notable losses involve VICReg (Bardes et al., 2021), Barlow Twins (Zbontar et al., 2021). The Barlow Twins loss is given as follows:

$$\sum_{i=1}^{B} (1 - \mathcal{C}_{ii})^2 + \lambda \sum_{i=1}^{B} \sum_{j \neq i} \mathcal{C}_{ij}^{2}, \quad (3)$$

where $\lambda$ is a hyperparameter and $\mathcal{C}_{ij}$ is the cross-correlation coefficient.

The idea of masked image modeling is to learn useful representations by generating the representation from partially visible patches and predicting the rest of the image from the representation, thus useful information in the image remains in the representation. We shall briefly introduce MAE (He et al., 2022) as an example. Given a batch of images $\{\mathbf{x}_i\}_{i=1}^{B}$, we shall first partition each of the images into $n$ disjoint patches $\mathbf{x}_i = \mathbf{x}_i(j)$ $(1 \leq j \leq n)$. Then $B$ random mask vectors $\mathbf{m}_i \in \{0,1\}^n$ will be generated, and denote the two images generated by these masks as

$$\mathbf{x}_i^{(1)} = \mathbf{x}_i \odot \mathbf{m}_i \quad \text{and} \quad \mathbf{x}_i^{(2)} = \mathbf{x}_i \odot (1 - \mathbf{m}_i). \quad (4)$$

The model consists of two modules: an encoder $f$ and a decoder $g$. The encoder transform each view $\mathbf{x}_i^{(1)}$ into a representation $\mathbf{z}_i = f(\mathbf{x}_i^{(1)})$. The loss function is $\sum_{i=1}^{B} \parallel g(\mathbf{z}_i) - \mathbf{x}^{(2)} \parallel_2^2$. We also denote the representations in a batch as $\mathbf{Z} = [\mathbf{z}_1, \cdots, \mathbf{z}_B]$.

The goal of this paper is to use matrix information maximization viewpoint to understand the seemingly different losses in contrastive and Feature Decorrelation based methods. We would like also to use matrix information-theoretic tools to improve MAE. We only analyze 3 popular losses: spectral contrastive, Barlow Twins and MAE.

## 3 APPLYING MATRIX INFORMATION THEORY TO CONTRASTIVE AND FEATURE DECORRELATION BASED METHODS

As we have discussed in the preliminary session, in contrastive and feature decorrelation based methods, a common practice is to use two branches (Siamese architecture) namely an online network and a target network to learn useful representations. However, the relationship of the two branches during the training process is mysterious. In this section, we shall use matrix information quantities to unveil the complicated relationship in Siamese architectures.

### 3.1 MEASURING THE MUTUAL INFORMATION

One interesting derivation in Oord et al. (2018) is that it can be shown that

$$\mathcal{L}_{\text{InfoNCE}} \geq -\operatorname{I}(\mathbf{Z}^{(1)}; \mathbf{Z}^{(2)}) + \log B, \tag{5}$$

where $\mathbf{Z}^{(i)}$ denotes the sampled distribution of the representation.

Though InfoNCE loss is a promising surrogate for estimating the mutual information between the two branches in self-supervised learning. Sordoni et al. (2021) doubt its effectiveness when facing high-dimensional inputs, where the mutual information may be larger than $\log B$, making the bound vacuous. Then a natural question arises: Can we calculate the mutual information exactly? Unfortunately, it is hard to calculate the mutual information reliably and effectively. Thus we may change our strategy by using the **matrix** mutual information instead.

As the matrix mutual information has an **exact** and easy-to-calculate expression, one question remains: How to choose the matrices used in the (matrix) mutual information? We find the (batch normalized) sample covariance matrix serves as a good candidate. The reason is that by using batch normalization, the empirical covariance matrix naturally satisfies the requirements that: All the diagonals equal to 1, the matrix is positive semi-definite and it is easy to estimate from data samples.

One vital problem with the bound Eqn. (5) is that it only provides an inequality, and thus doesn't directly show the **exact** equivalence of contrastive learning and mutual information maximization. In the following, we will show that two pivotal self-supervised learning methods are exactly maximizing the mutual information. Specifically, we consider setting the $\alpha$ in entropy to be 2.

We shall first present a proposition that relates the mutual information with the Frobenius norm.

**Proposition 3.1.** $\operatorname{I}_2(\mathbf{K}_1; \mathbf{K}_2) = 2\log d - \log \frac{||\mathbf{K}_1||_F^2 ||\mathbf{K}_2||_F^2}{||\mathbf{K}_1 \odot \mathbf{K}_2||_F^2}$, where $d$ is the size of matrix $\mathbf{K}_1$.

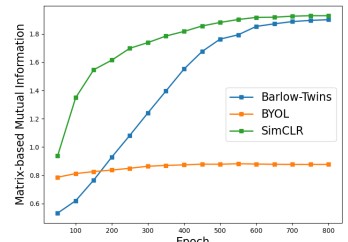

Figure 1: Visualization of matrix-based mutual information on CIFAR10 for Barlow-Twins, BYOL and Sim-CLR.

One thing interesting in matrix information theory that distinguishes it from traditional information theory is that it can not only deal with samples from batches but also can exploit the relationship among batches, which we will informally call batch-dimension duality. Specifically, the sample covariance matrix can be expressed as $\mathbf{Z}\mathbf{Z}^\top \in \mathbb{R}^{d \times d}$ and the batch-sample Gram matrix can be expressed as $\mathbf{Z}^\top \mathbf{Z} \in \mathbb{R}^{B \times B}$. The closeness of these two matrices makes us call $B$ and $d$ has duality.

Notably, spectral contrastive loss (HaoChen et al., 2021) is a good surrogate loss for InfoNCE loss and calculates the loss involving the batch Gram matrix. Another famous loss used in feature decorrelation based methods is the Barlow Twins (Zbontar et al., 2021), which involves the batch-normalized sample covariance matrix. As we have discussed earlier, these two losses can be seen as a natural

duality pair. In the following, we shall prove that these two losses **exactly** maximize the (matrix) mutual information at the optimal point.

**Theorem 3.2.** *The optimal point of Barlow twins and spectral contrastive learning losses maximize the matrix mutual information.*

*Proof.* Please refer to Appendix A. □

What about the mutual information when $\alpha = 1$? We shall then plot the mutual information of covariance matrices between branches in Figure 1. We can find out that the mutual information increases during training, which is similar to the case of $\alpha = 2$ proved by Theorem 3.2. More interestingly, the mutual information of SimCLR and Barlow Twins meet at the end of training, strongly emphasizing the duality of these algorithms.

### 3.2 MEASURING THE (JOINT) ENTROPY

After discussing the application of matrix mutual information in self-supervised learning. We wonder how the entropy evolves during the process.

Denote $\mathbf{K}_1 = \mathbf{Z}_1\mathbf{Z}_1^\top$ and $\mathbf{K}_2 = \mathbf{Z}_2\mathbf{Z}_2^\top$ for the online network and target network respectively. We can show that the matrix joint entropy can indeed reflect the dimensions of representations in Siamese architectures.

**Proposition 3.3.** *The joint entropy lower bounds the representation rank in two branches by having the inequality as follows:*

$$\mathrm{H}_1(\mathbf{K}_1, \mathbf{K}_2) \leq \log(\mathrm{rank}(\mathbf{K}_1 \odot \mathbf{K}_2)) \leq \log \mathrm{rank}(\mathbf{K}_1) + \log \mathrm{rank}(\mathbf{K}_2). \tag{6}$$

This proposition shows that the bigger the joint entropy between the two branches is, the less likely that the representation collapse. We shall also introduce another surrogate for entropy estimation which is closely related to matrix entropy:

**Definition 3.4.** Suppose $B$ samples $\mathbf{Z} = [\mathbf{z}_1, \mathbf{z}_2, \cdots, \mathbf{z}_B] \in \mathbb{R}^{d \times B}$ are i.i.d. samples from a distribution $p(z)$. Then the total coding rate (TCR) (Yu et al., 2020) of $p(z)$ is defined as follows:

$$\mathrm{TCR}_\mu(\mathbf{Z}) = \log \det(\mu \mathbf{I}_d + \mathbf{Z}\mathbf{Z}^\top), \tag{7}$$

where $\mu$ is a non-negative hyperparameter.

For notation simplicity, we shall also write $\mathrm{TCR}_\mu(\mathbf{Z})$ as $\mathrm{TCR}_\mu(\mathbf{Z}\mathbf{Z}^\top)$. We shall then show the close relationship between TCR and matrix entropy in the following theorem through the lens of matrix KL divergence. The key is utilizing the asymmetries of the matrix KL divergence.

**Proposition 3.5.** *Suppose $\mathbf{K}$ is a $d \times d$ matrix with the constraint that each of its diagonals is $1$. Then the following equalities holds:*

$$\mathrm{H}_1(\mathbf{K}) = \log d - \frac{1}{d}\mathrm{KL}(\mathbf{K}, \mathbf{I}_d), \text{ and } \mathrm{TCR}_\mu(\mathbf{K}) = d\log(1+\mu) - \mathrm{KL}(\mathbf{I}_d, \frac{1}{1+\mu}(\mu\mathbf{I}_d + \mathbf{K})). \tag{8}$$

As TCR can be treated as a good surrogate for entropy, we can obtain the following bound.

**Proposition 3.6.** *The (joint) total coding rate upperbounds the rate in two branches by having the inequality as follows:*

$$\mathrm{TCR}_{\mu^2+2\mu}(\mathbf{K}_1 \odot \mathbf{K}_2) \geq \mathrm{TCR}_\mu(\mathbf{K}_1) + \mathrm{TCR}_\mu(\mathbf{K}_2). \tag{9}$$

Combining Propositions 3.5, 3.3, and 3.6, it is clear that by using TCR as a surrogate for entropy, the bigger the entropy is for each branch the bigger the joint entropy. Thus by combining the conclusion from the above two theorems, it is evident that the joint entropy strongly reflects the extent of collapse during training.

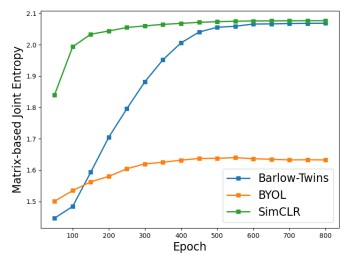

Figure 2: Visualization of matrix-based joint entropy on CIFAR10 for Barlow-Twins, BYOL and SimCLR.

What about the joint entropy when $\alpha = 1$ behaves empirically? We shall then plot the joint entropy of covariance matrices between branches in Figure 2. We can find out that the joint entropy increases during training. More interestingly, the joint entropy of SimCLR and Barlow Twins meet at the end of training, strongly reflects a duality of these algorithms. Notably, we can also show that spectral contrastive and Barlow Twins maximize **exactly** the joint entropy between branches during training. This remarkable conclusion is proved when the Renyi entropy order $\alpha = 2$.

We shall first present a proposition that relates the joint entropy with the Frobenius norm.

**Proposition 3.7.** *Suppose* $\mathbf{K}_1, \mathbf{K}_2 \in \mathbb{R}^{d \times d}$. *Then* $\mathrm{H}_2(\mathbf{K}_1, \mathbf{K}_2) = 2\log d - \log || \mathbf{K}_1 \odot \mathbf{K}_2 ||_F^2$, *where $F$ is the Frobenius norm.*

Then we can show that the optimal point of spectral contrastive loss and Barlow twins loss can be seen as maximizing the joint entropy between branches.

**Theorem 3.8.** *The optimal point of Barlow twins and spectral contrastive learning losses maximize the matrix joint entropy.*

*Proof.* Please refer to Appendix A. □

## 4 Applying Matrix Information Theory to Masked Image Modeling

As contrastive and feature decorrelation based methods usually surpass MAE by a large margin on linear probing. One may wonder if can we apply this matrix information theory to improve MAE. From a traditional information-theoretic point of view, when the two branches merge into one branch the mutual information $\mathrm{I}(\mathbf{X}; \mathbf{X})$ and the joint entropy $\mathrm{H}(\mathbf{X}, \mathbf{X})$ both equal to the Shannon entropy $\mathrm{H}(\mathbf{X})$. Thus we would like to use the matrix entropy in MAE training. Moreover, matrix entropy can be shown to be very close to a quantity called effective rank. And Zhang et al. (2022b); Garrido et al. (2023) show that the effective rank is a critical quantity for better representation. The definition of effective rank is formally stated in Definition 4.1 and it is easy to show when the matrix is positive semi-definite and has all its diagonal being 1 the effective rank is the exponential of the matrix entropy.

**Definition 4.1** (Effective Rank (Roy & Vetterli, 2007)). For a non-all-zero matrix $\mathbf{A} \in \mathbb{C}^{n \times n}$, the effective rank, denoted as $\mathrm{erank}(\mathbf{A})$, is defined as

$$\mathrm{erank}(\mathbf{A}) \triangleq \exp\left(\mathrm{H}\left(p_1, p_2, \ldots, p_n\right)\right), \tag{10}$$

where $p_i = \frac{\sigma_i}{\sum_{k=1}^n \sigma_k}$, $\{\sigma_i \mid i = 1, \cdots, n\}$ represents the singular values of $\mathbf{A}$, and $\mathrm{H}$ denotes the Shannon entropy.

Thus it is natural to add the matrix entropy to the MAE loss to give a new self-supervised learning method. As the numerical instability of calculating matrix entropy is larger than its proxy TCR, we shall use TCR loss instead. One may wonder if we can link this new loss to other traditional unsupervised learning methods to give a better understanding. We answer this in the affirmative by linking this to the VAE approaches.

Recall the loss for traditional variational auto-encoder which is given as follows.

$$\mathcal{L}_{\mathrm{VAE}} = \mathbb{E}_{\mathbf{x} \sim \tilde{p}(\mathbf{x})}[-\log q(\mathbf{x} \mid \mathbf{z}) + \mathrm{KL}(p(\mathbf{z} \mid \mathbf{x})\|q(\mathbf{z}))], \quad \mathbf{z} \sim p(\mathbf{z} \mid \mathbf{x}).$$

The loss contains two terms, the first term $-\log q(\mathbf{x} \mid \mathbf{z})$ is a reconstruction loss that measures the decoding accuracy. The second term is a discriminative term, which measures the divergence of the encoder distribution $p(\mathbf{z} \mid \mathbf{x})$ with the latent distribution $q(\mathbf{z})$.

In the context of masked image modeling, we usually use MSE loss in place of the log-likelihood. For any input image $\mathbf{x}$, the process of randomly generating a masked vector $m$ and obtaining $\mathbf{z} = f(\mathbf{x} \odot \mathbf{m})$ can be seen as modeling the generating process of $\mathbf{z} \mid \mathbf{x}$. The decoding process $\mathbf{x} \mid \mathbf{z}$ can be modeled by concatenating $g(\mathbf{z})$ and $\mathbf{x} \odot \mathbf{m}$ by the (random) position induced by $m$. Thus the reconstruction loss will be $|| \mathrm{concat}(g(\mathbf{z}), \mathbf{x} \odot \mathbf{m}) - \mathbf{x} ||_2^2 = || g(\mathbf{z}) - \mathbf{x} \odot (1 - \mathbf{m}) ||_2^2$. For a batch of images $\{\mathbf{x}_i\}_{i=1}^B$, this exactly recovers the MAE loss.

Recall that we assume each representation $z_i$ is $l_2$ normalized. If we take the latent distribution of $Z$ as the uniform distribution on the unit hyper-sphere $S^{d-1}$, we shall get the following matrix variational masked auto-encoder (M-MAE) loss for self-supervised learning.

$$\mathcal{L}_{\text{M-MAE}} \triangleq \mathcal{L}_{\text{MAE}} - \lambda \cdot \text{TCR}_\mu(\mathbf{Z}), \tag{11}$$

where $\lambda$ is a loss-balancing hyperparameter.

Very interestingly, we can show that U-MAE is a second-order approximation of our proposed M-MAE. The key point is noticing that representations are $l_2$ normalized and using Taylor expansion as follows:

$$\begin{aligned}
\mathcal{L}_{\text{M-MAE}} &= \mathcal{L}_{\text{MAE}} - \lambda \log \det(\mathbf{I}_d + \frac{1}{\mu} \mathbf{Z}\mathbf{Z}^\top) + \text{Const.} \\
&= \mathcal{L}_{\text{MAE}} - \lambda \log \det(\mathbf{I}_B + \frac{1}{\mu} \mathbf{Z}^\top \mathbf{Z}) + \text{Const.} \\
&= \mathcal{L}_{\text{MAE}} - \lambda \operatorname{tr} \log(\mathbf{I}_B + \frac{1}{\mu} \mathbf{Z}^\top \mathbf{Z}) + \text{Const.} \\
&= \mathcal{L}_{\text{MAE}} - \lambda \operatorname{tr}(\frac{1}{\mu} \mathbf{Z}^\top \mathbf{Z} - \frac{1}{2\mu^2} (\mathbf{Z}^\top \mathbf{Z})^2 + \cdots) \\
&= \mathcal{L}_{\text{U-MAE}} + \text{Higher-order-terms} + \text{Const.}
\end{aligned}$$

## 5 EXPERIMENTS

In this section, we rigorously evaluate our Matrix Variational Masked Auto-Encoder (M-MAE) with TCR loss, placing special emphasis on its performance in comparison to the U-MAE model with Square uniformity loss as a baseline. This experiment aims to shed light on the benefits that matrix information-theoretic tools can bring to self-supervised learning algorithms.

### 5.1 EXPERIMENTAL SETUP

**Datasets: ImageNet-1K.** We utilize the ImageNet-1K dataset (Deng et al., 2009), which is one of the most comprehensive datasets for image classification. It contains over 1 million images spread across 1000 classes, providing a robust platform for evaluating our method's generalization capabilities.

**Model Architectures.** We adopt Vision Transformers (ViT) such as ViT-Base and ViT-Large for our models, following the precedent settings by the U-MAE (Zhang et al., 2022b) paper.

**Hyperparameters.** For a fair comparison, we adopt U-MAE's original hyperparameters: a mask ratio of 0.75 and a uniformity term coefficient $\lambda$ of 0.01 by default. Both models are pre-trained for 200 epochs on ImageNet-1K with a batch size of 1024, and weight decay is similarly configured as 0.05 to ensure parity in the experimental conditions. For ViT-Base, we set the TCR coefficients $\mu = 1$, and for ViT-Large, we set $\mu = 3$.

### 5.2 EVALUATION RESULTS

**Evaluation Metrics.** From Table 1, it's evident that the M-MAE loss significantly outperforms both MAE and U-MAE in terms of linear evaluation and fine-tuning accuracy. Specifically, for ViT-Base, M-MAE achieves a linear probing accuracy of 62.4%, which is a substantial improvement over MAE's 55.4% and U-MAE's 58.5%. Similarly, in the context of ViT-Large, M-MAE achieves an accuracy of 66.0%, again surpassing both MAE and U-MAE. In terms of fine-tuning performance, M-MAE also exhibits superiority, achieving 83.1% and 84.3% accuracy for ViT-Base and ViT-Large respectively. Notably, a 1% increase in accuracy at ViT-Large is very significant. These results empirically validate the theoretical advantages of incorporating matrix information-theoretic tools into self-supervised learning, as encapsulated by the TCR loss term in the M-MAE loss function.

To investigate the robustness of our approach to variations in hyperparameters, we perform an ablation study focusing on the coefficients $\mu$ in the TCR loss. The results for different $\mu$ values are summarized as follows:

Table 1: Linear evaluation accuracy (%) and fine-tuning accuracy (%) of pretrained models by MAE loss, U-MAE loss, and M-MAE loss with different ViT backbones on ImageNet-1K. The uniformity regularizer TCR loss in the M-MAE loss significantly improves the linear evaluation performance and fine-tuning performance of the MAE loss.

| Downstream Task | Method | ViT-Base | ViT-Large |
|---|---|---|---|
| Linear Probing | MAE | 55.4 | 62.2 |
| | U-MAE | 58.5 | 65.8 |
| | M-MAE | **62.4** | **66.0** |
| Fine-tuning | MAE | 82.9 | 83.3 |
| | U-MAE | 83.0 | 83.2 |
| | M-MAE | **83.1** | **84.3** |

Table 2: Linear probing accuracy (%) of M-MAE for ViT-Base with varying $\mu$ coefficients.

| $\mu$ Coefficient | 0.1 | 0.5 | 0.75 | 1 | 1.25 | 1.5 | 3 |
|---|---|---|---|---|---|---|---|
| Accuracy | 58.61 | 59.38 | 59.87 | 62.40 | 59.54 | 57.76 | 50.46 |

As observed in Table 2, the M-MAE model exhibits a peak performance at $\mu = 1$ for ViT-Base. Deviating from this value leads to a gradual degradation in performance, illustrating the importance of careful hyperparameter tuning for maximizing the benefits of the TCR loss.

## 6 RELATED WORK

**Self-supervised learning.** Contrastive and feature decorrelation based methods have emerged as powerful approaches for unsupervised representation learning. These methods offer an alternative paradigm to traditional supervised learning, eliminating the reliance on human-annotated labels. By leveraging diverse views or augmentations of input data, they aim to capture meaningful and informative representations that can generalize across different tasks and domains (Chen et al., 2020; Hjelm et al., 2018; Wu et al., 2018; Tian et al., 2019; Chen & He, 2021; Gao et al., 2021; Bachman et al., 2019; Oord et al., 2018; Ye et al., 2019; Henaff, 2020; Misra & Maaten, 2020; Caron et al., 2020; HaoChen et al., 2021; Caron et al., 2021; Li et al., 2021; Zbontar et al., 2021; Tsai et al., 2021; Bardes et al., 2021; Tian et al., 2020; Robinson et al., 2021; Dubois et al., 2022). These representations can be used for various of downstream tasks, achieving remarkable performance and even outperforming supervised feature representations. These self-supervised learning methods have the potential to unlock the latent information present in unlabeled data, enabling the development of more robust and versatile models in various domains.

In recent times, Masked Image Modeling (MIM) (Zhang et al., 2022a) has gained attention as a visual representation learning approach inspired by the widely adopted Masked Language Modeling (MLM) paradigm in NLP, such as exemplified by BERT (Devlin et al., 2018). Notably, several MIM methods, including iBOT Zhou et al. (2021), SimMIM (Xie et al., 2022), and MAE (He et al., 2022), have demonstrated promising results in this domain. While iBOT and SimMIM share similarities with BERT and have direct connections to contrastive learning, MAE diverges from BERT-like methods. MAE, aptly named, deviates from being solely a token predictor and leans towards the autoencoder paradigm. It distinguishes itself by employing a pixel-level reconstruction loss, excluding the masked tokens from the encoder input, and employing a fully non-linear encoder-decoder architecture. This unique design allows MAE to capture intricate details and spatial information within the input images.

**Matrix information theory.** Information theory that provides a framework for understanding the relationship between probability and information. Recently, apart from traditional information theory which usually involves calculating information-theoretic quantities on sample distributions. Recently, there have been attempts to generalize information theory to measure the relationships between matrices (Bach, 2022; Skean et al., 2023; Zhang et al., 2023b;a). The idea is to apply the traditional information-theoretic quantities on the spectrum of matrices. Compared to (traditional) information

theory, matrix information theory can be seen as measuring the "second-order" relationship between distributions (Zhang et al., 2023b).

**Theoretical understanding of self-supervised learning.** The practical achievements of contrastive learning have ignited a surge of theoretical investigations into the understanding how contrastive loss works Arora et al. (2019); HaoChen et al. (2021; 2022); Tosh et al. (2020; 2021); Lee et al. (2020); Wang et al. (2022); Nozawa & Sato (2021); Huang et al. (2021); Tian (2022); Hu et al. (2022); Tan et al. (2023). Wang & Isola (2020) provide an insightful analysis of the optimal solutions of the InfoNCE loss, providing insights into the alignment term and uniformity term that constitute the loss, thus contributing to a deeper understanding of self-supervised learning. HaoChen et al. (2021); Wang et al. (2022); Tan et al. (2023) explore contrastive self-supervised learning methods from a spectral graph perspective. In addition to these interpretations of contrastive losses, Saunshi et al. (2022); HaoChen & Ma (2022) find that inductive bias plays a pivotal and influential role in shaping the downstream performance of self-supervised learning. In their seminal work, Cabannes et al. (2023) present a comprehensive theoretical framework that sheds light on the intricate interplay between the selection of data augmentation techniques, the network architectures, and the choice of training algorithms.

Several theoretical investigations have delved into the realm of feature decorrelation based methods within the domain of self-supervised learning, as evidenced by a collection of notable studies (Wen & Li, 2022; Tian et al., 2021; Garrido et al., 2022; Balestriero & LeCun, 2022; Tsai et al., 2021; Pokle et al., 2022; Tao et al., 2022; Lee et al., 2021). Balestriero & LeCun (2022) have unveiled intriguing connections between variants of SimCLR, Barlow Twins, and VICReg, and classical unsupervised learning techniques. The resilience of methods like SimSiam against collapse has been a subject of investigation, as analyzed by Tian et al. (2021). Pokle et al. (2022) have undertaken a comparative exploration of the loss landscape between SimSiam and SimCLR, thereby revealing the presence of suboptimal minima in the latter. In another study, (Tsai et al., 2021) have established a connection between a variant of the Barlow Twins' criterion and a variant of the Hilbert-Schmidt Independence Criterion (HSIC). In addition to these findings, the theoretical aspects of data augmentation in sample-contrastive learning have been thoroughly examined by (Huang et al., 2021; Wen & Li, 2021), adding further depth to the understanding of this area of research.

Compared to contrastive and feature decorrelation based methods, the theoretical understanding of masked image modeling is still in an early stage. Cao et al. (2022) use the viewpoint of the integral kernel to understand MAE. Zhang et al. (2022b) use the idea of a masked graph to relate MAE with the alignment loss in contrastive learning. Recently, Kong et al. (2023) show MAE effectively detects and identifies a specific group of latent variables using a hierarchical model.

## 7 CONCLUSION

In conclusion, this study delves into self-supervised learning (SSL), examining contrastive, non-contrastive learning, and masked image modeling through the lens of matrix information theory. Our exploration reveals that many SSL methods are maximizing matrix information-theoretic quantities on Siamese architectures at their optimal point. We also introduce a novel method, the matrix variational masked auto-encoder (M-MAE), enhancing masked image modeling by adding matrix entropy. This not only deepens our understanding of existing SSL methods but also propels performance in linear probing and fine-tuning tasks to surpass state-of-the-art metrics.

The insights underscore matrix information theory's potency in analyzing and refining SSL methods, paving the way towards a unified toolbox for advancing SSL, irrespective of the methodological approach. This contribution augments the theoretical and practical landscape of SSL, hoping to spark further research and innovations in SSL. Future endeavors could explore refining SSL methods on Siamese architectures and advancing masked image modeling methods using matrix information theory tools, like new estimators for matrix entropy.

## REPRODUCIBILITY STATEMENT

To foster reproducibility, we submit our experiment code as supplementary material. One can directly reproduce the experiment results following the instructions in the README document. We also give experiment details in Section 5.

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

# A APPENDIX FOR PROOFS

**Proof of Propositions 3.1 and 3.7**

*Proof.* The proof is straightforward by using the definition of matrix mutual information when $\alpha = 2$ and the fact that when $\mathbf{K}$ is symmetric $\text{tr}(\mathbf{K}^2) = \text{tr}(\mathbf{K}^T\mathbf{K}) = \| \mathbf{K} \|_F^2$. $\qquad\square$

**Proof of Theorem 3.2.** We shall first present a lemma as follows:

**Lemma A.1.** *Given two positive integers $n, m$. Denote two sequences $\mathbf{x} = (x_1, \cdots, x_m)$ and $\mathbf{y} = (y_1, \cdots, y_m)$. Then $\mathbf{x} = \mathbf{y} = 0$ is the unique solution to the following optimization problem:*

$$\min_{0 \leq \mathbf{x} \leq 1, 0 \leq \mathbf{y} \leq 1} \frac{(n + \sum_{i=1}^m x_i)(n + \sum_{i=1}^m y_i)}{n + \sum_{i=1}^m x_i y_i}.$$

*Proof.* Notice that

$$\frac{(n + \sum_{i=1}^m x_i)(n + \sum_{i=1}^m y_i)}{n + \sum_{i=1}^m x_i y_i} - n = \frac{n(\sum_{i=1}^m x_i + \sum_{i=1}^m y_i) - n \sum_{i=1}^m x_i y_i + (\sum_{i=1}^m x_i)(\sum_{i=1}^m y_i)}{n + \sum_{i=1}^m x_i y_i}$$

Note $x_i \geq x_i^2$ and $y_i \geq y_i^2$. Then we shall get inequality as follows:

$$\sum_{i=1}^m x_i + \sum_{i=1}^m y_i \geq 2\sqrt{(\sum_{i=1}^m x_i)(\sum_{i=1}^m y_i)} \geq 2\sqrt{(\sum_{i=1}^m x_i^2)(\sum_{i=1}^m y_i^2)} \geq 2\sum_{i=1}^m x_i y_i.$$

Thus the above optimization problem gets a minimum of $n$, with $\mathbf{x} = \mathbf{y} = 0$ the unique solution. $\quad\square$

*Proof.* Denote the (batch normalized) vectors for each dimension $i$ ($1 \leq i \leq d$) of the online and target networks as $\bar{\mathbf{z}}_i^{(1)}$ and $\bar{\mathbf{z}}_i^{(2)}$.

Take $\mathbf{K}_1 = [\bar{\mathbf{z}}_1^{(1)} \cdots \bar{\mathbf{z}}_d^{(1)}]^\top [\bar{\mathbf{z}}_1^{(1)} \cdots \bar{\mathbf{z}}_d^{(1)}]$ and $\mathbf{K}_2 = [\bar{\mathbf{z}}_1^{(2)} \cdots \bar{\mathbf{z}}_d^{(2)}]^\top [\bar{\mathbf{z}}_1^{(2)} \cdots \bar{\mathbf{z}}_d^{(2)}]$.

From Proposition 3.1, it is clear that the mutual information $\mathbf{I}_2(\mathbf{K}_1; \mathbf{K}_2)$ is maximized iff $\frac{\|\mathbf{K}_1\|_F^2 \|\mathbf{K}_2\|_F^2}{\|\mathbf{K}_1 \odot \mathbf{K}_2\|_F^2}$ is minimized. Take $((\bar{\mathbf{z}}_i^{(1)})^\top \bar{\mathbf{z}}_j^{(1)})^2$ and $((\bar{\mathbf{z}}_i^{(2)})^\top \bar{\mathbf{z}}_j^{(2)})^2$ as elements of $\mathbf{x}$ and $\mathbf{y}$ in Lemma A.1, then we can see the maximal mutual information is attained iff $(\bar{\mathbf{z}}_i^{(1)})^\top \bar{\mathbf{z}}_j^{(1)} = 0$ and $(\bar{\mathbf{z}}_i^{(2)})^\top \bar{\mathbf{z}}_j^{(2)} = 0$.

As the optimal point of Barlow Twins loss has $\bar{\mathbf{z}}_i^{(1)} = \bar{\mathbf{z}}_i^{(2)}$ for each $i \in \{1, \cdots, d\}$ and $(\bar{\mathbf{z}}_i^{(1)})^\top \bar{\mathbf{z}}_j^{(2)} = 0$ for each $i \neq j$. Then for each $i \neq j$, $(\bar{\mathbf{z}}_i^{(1)})^\top \bar{\mathbf{z}}_j^{(1)} = (\bar{\mathbf{z}}_i^{(1)})^\top \bar{\mathbf{z}}_j^{(2)} = 0$. Similarly, $(\bar{\mathbf{z}}_i^{(2)})^\top \bar{\mathbf{z}}_j^{(2)} = 0$. Then the matrix mutual information is maximized.

When performing spectral contrastive learning, the loss is $\sum_{i=1}^B \| \mathbf{z}_i^{(1)} - \mathbf{z}_i^{(2)} \|_2^2 + \lambda \sum_{i \neq j} ((\mathbf{z}_i^{(1)})^\top \mathbf{z}_j^{(2)})^2$. Take $\mathbf{K}_1 = \mathbf{Z}_1^T \mathbf{Z}_1$ and $\mathbf{K}_2 = \mathbf{Z}_2^T \mathbf{Z}_2$, the results follows similarly. Thus concludes the proof. $\qquad\square$

**Proof of Proposition 3.3.**

*Proof.* The first inequality comes from the fact that effective rank lower bounds the rank. The second inequality comes from the rank inequality of Hadamard product. $\qquad\square$

**Proof of Proposition 3.5.**

*Proof.* The proof is from directly using the definition of matrix KL divergence. $\qquad\square$

**Proof of Proposition 3.6.**

*Proof.* The inequality comes from the determinant inequality of Hadamard products and the fact that $(\mathbf{K}_1 + \mu\mathbf{I}) \odot (\mathbf{K}_1 + \mu\mathbf{I}) = \mathbf{K}_1 \odot \mathbf{K}_2 + (\mu^2 + 2\mu)\mathbf{I}$. $\qquad\square$

**Proof of Theorem 3.8.**

*Proof.* Denote the (along batch normalized) vectors for each dimension $i$ ($1 \leq i \leq d$) of the online and target networks as $\bar{\mathbf{z}}_i^{(1)}$ and $\bar{\mathbf{z}}_i^{(2)}$. Take $\mathbf{K}_1 = [\bar{\mathbf{z}}_1^{(1)} \cdots \bar{\mathbf{z}}_d^{(1)}]^\top [\bar{\mathbf{z}}_1^{(1)} \cdots \bar{\mathbf{z}}_d^{(1)}]$ and $\mathbf{K}_2 = [\bar{\mathbf{z}}_1^{(2)} \cdots \bar{\mathbf{z}}_d^{(2)}]^\top [\bar{\mathbf{z}}_1^{(2)} \cdots \bar{\mathbf{z}}_d^{(2)}]$. From Proposition 3.7, it is clear that the joint entropy $\mathrm{H}_2(\mathbf{K}_1, \mathbf{K}_2)$ is maximized iff $\| \mathbf{K}_1 \odot \mathbf{K}_2 \|_F^2$ is minimized. Note from the definition of Frobenius norm, $\| \mathbf{K}_1 \odot \mathbf{K}_2 \|_F^2 = \sum_{i,j}((\mathbf{K}_1 \odot \mathbf{K}_2)(i,j))^2 = \sum_{i,j}(\mathbf{K}_1(i,j)\mathbf{K}_2(i,j))^2$. As the optimal point of of Barlow Twins loss has $\bar{\mathbf{z}}_i^{(1)} = \bar{\mathbf{z}}_i^{(2)}$ for each $i \in \{1, \cdots, d\}$ and $(\bar{\mathbf{z}}_i^{(1)})^\top \bar{\mathbf{z}}_j^{(2)} = 0$ for each $i \neq j$. Then for each $i \neq j$, $(\bar{\mathbf{z}}_i^{(1)})^\top \bar{\mathbf{z}}_j^{(1)} = (\bar{\mathbf{z}}_i^{(1)})^\top \bar{\mathbf{z}}_j^{(2)} = 0$. Similarly, $(\bar{\mathbf{z}}_i^{(2)})^\top \bar{\mathbf{z}}_j^{(2)} = 0$. When performing spectral contrastive learning, the loss is $\sum_{i=1}^B \| \mathbf{z}_i^{(1)} - \mathbf{z}_i^{(2)} \|_2^2 + \lambda \sum_{i \neq j}((\mathbf{z}_i^{(1)})^\top \mathbf{z}_j^{(2)})^2$. Take $\mathbf{K}_1 = \mathbf{Z}_1^T \mathbf{Z}_1$ and $\mathbf{K}_2 = \mathbf{Z}_2^T \mathbf{Z}_2$, the results follows similarly. $\square$

Remark: Following the proof of our Theorems 3.2 and 3.8, our theoretical results can be generalized to sample contrastive and dimension contrastive methods defined in (Garrido et al., 2022). As pointed out by (Garrido et al., 2022), sample and dimension contrastive methods contain many famous self-supervised methods (Proposition 3.2 of (Garrido et al., 2022)).

## B  MEASURING THE DIFFERENCE BETWEEN SIAMESE BRANCHES

As we have discussed the total or shared information in the Siamese architectures, we haven't used the matrix information-theoretic tools to analyze the **differences** in the two branches.

From information theory, we know that KL divergence is a special case of $f$-divergence defined as follows:

**Definition B.1.** For two probability distributions $\mathbf{P}$ and $\mathbf{Q}$, where $\mathbf{P}$ is absolutely continuous with respect to $\mathbf{Q}$. Suppose $\mathbf{P}$ and $\mathbf{Q}$ has density $p(x)$ and $q(x)$ respectively. Then for a convex function $f$ is defined on non-negative numbers which is right-continuous at $0$ and satisfies $f(1) = 0$. The $f$-divergence is defined as:

$$D_f(\mathbf{P} \| \mathbf{Q}) = \int f\left(\frac{p(x)}{q(x)}\right) q(x)dx. \tag{12}$$

When $f(x) = x \log x$ will recover the KL divergence. Then a natural question arises: are there other $f$ divergences that can be easily generalized to matrices? Note by taking $f(x) = -(x + 1) \log \frac{x+1}{2} + x \log x$, we shall retrieve JS divergence. Recently, Hoyos-Osorio & Sanchez-Giraldo (2023) generalized JS divergence to the matrix regime.

**Definition B.2** (Matrix JS divergence (Hoyos-Osorio & Sanchez-Giraldo, 2023))**.** Suppose matrix $\mathbf{K}_1, \mathbf{K}_2 \in \mathbb{R}^{n \times n}$ which $\mathbf{K}_1(i,i) = \mathbf{K}_2(i,i) = 1$ for every $i = 1, \cdots, n$. The Jensen-Shannon (JS) divergence between these two matrices $\mathbf{K}_1$ and $\mathbf{K}_2$ is defined as

$$\mathrm{JS}\left(\mathbf{K}_1 \| \mathbf{K}_2\right) = \mathbf{H}_1\left(\frac{\mathbf{K}_1 + \mathbf{K}_2}{2}\right) - \frac{\mathbf{H}_1(\mathbf{K}_1) + \mathbf{H}_1(\mathbf{K}_2)}{2}.$$

One may think the matrix KL divergence is a good candidate, but this quantity has some severe problems making it not a good choice. One problem is that the matrix KL divergence is not symmetric. Another problem is that the matrix KL divergence is not bounded, and sometimes may even be undefined. Recall these drawbacks are similar to that of KL divergence in traditional information theory. In traditional information theory, JS divergence successfully overcomes these drawbacks, thus we may use the matrix JS divergence to measure the differences between branches. As matrix JS divergence considers the interactions between branches, we shall also include the JS divergence between eigenspace distributions as another difference measure.

Specifically, the online and target batch normalized feature correlation matrices can be calculated by $\mathbf{K}_1 = \mathbf{Z}_1 \mathbf{Z}_1^\top$ and $\mathbf{K}_2 = \mathbf{Z}_2 \mathbf{Z}_2^\top$. Denote $\mathbf{p}_1$ and $\mathbf{p}_2$ the online and target (normalized) eigen distribution respectively. We plot the matrix JS divergence $\mathrm{JS}(\mathbf{K}_1, \mathbf{K}_2)$ between branches in Figure 3(a). It is evident that throughout the whole training, the JS divergence is a small value, indicating a small gap

between the branches. More interestingly, the JS divergence increases during training, which means that an effect of "symmetry-breaking" may exist in self-supervised learning. Additionally, we plot the plain JS divergence $JS(\mathbf{p_1}, \mathbf{p_2})$ between branches in Figure 3(b). It is evident that $JS(\mathbf{p_1}, \mathbf{p_2})$ is very small, even compared to $JS(\mathbf{K_1}, \mathbf{K_2})$. Thus we hypothesize that the "symmetry-breaking" phenomenon is mainly due to the interactions between Siamese branches.

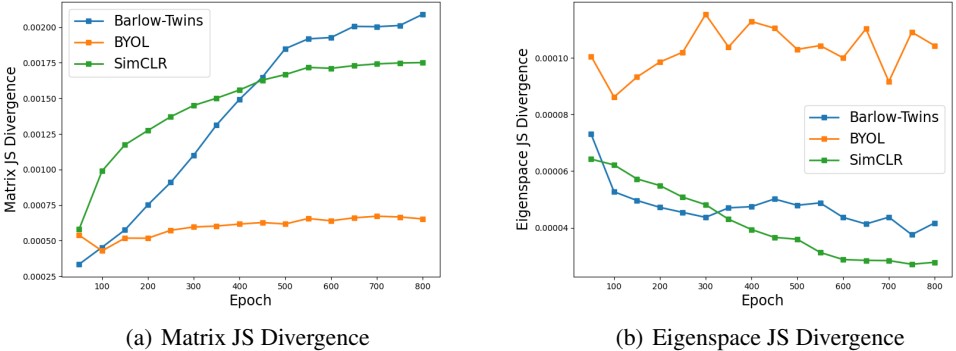

(a) Matrix JS Divergence          (b) Eigenspace JS Divergence

Figure 3: Visualization of matrix JS divergence and eigenspace JS divergence on CIFAR10 for Barlow-Twins, BYOL and SimCLR.

