# OpenReview forum: "Information Flow in Self-Supervised Learning"
_ICLR.cc/2024/Conference — Submitted to ICLR 2024_

### Official Review · Reviewer_j36B · 2023-10-30

**Soundness:** 2 fair
**Presentation:** 3 good
**Contribution:** 2 fair
**Rating:** 5
**Confidence:** 3

**Summary:**

This paper aims at providing better understanding for existing methods (contrastive and feature decorrelation based methods) by leveraging the principles of matrix mutual information and joint entropy. In addition, the paper proposes the "matrix variational masked auto-encoder" (M-MAE) method. The paper reports empircal results that show the effectiveness of M-MAE compared with the state-of-the-art methods for representation learning on ImageNet.

**Strengths:**

- The paper tackles important questions in the field of self-supervised learning

**Weaknesses:**

- The proofs are not always clear/complete (see questions below).
- There are propositions and theorems in the paper. Then in the appendix, there are only proofs to "theorems" and the reader must match correct theorem/proposition with the correct proof.
- The experimental setup lacks many details.

**Questions:**

- Theorems 1 and 2 do not seem to be proved clearly. The proof to theorem 1 in the appendix (the one that includes "lemma 1"!) does not seem to proof things by following a clear mathematical reasoning. Can you clarify how the last sentences leads to a valid proof? It it the same for Theorem 2 (the proof on page 15 given the other one seems to refer to Proposition 2).
- Can you provide additional technical details about the experimental setup. The appendix does not seem to contain any information related to that and only limited information is given in the main paper (what are exactly the training objectives and hyper-parameters such as batch size, etc.).
- It is not fully clear how Theorems 1 and 2 are not going against each other given Definition 2. Can you provide some information about that?

---

> ### Author Response · Authors · 2023-11-18
> **Response to reviewer j36B**
>
> >Q1: Theorems 1 and 2 do not seem to be proved clearly. The proof to theorem 1 in the appendix (the one that includes "lemma 1"!) does not seem to proof things by following a clear mathematical reasoning. Can you clarify how the last sentences leads to a valid proof? It it the same for Theorem 2 (the proof on page 15 given the other one seems to refer to Proposition 2).
>
> A1: Thank you for your feedback. **We stated in the general response that "the last sentence" $K_1 = Z_1$ and $K_2 = Z_2$ is a typo and we have corrected it to $K_1=Z^T_1Z_1$ and $K_2=Z^T_2Z_2$ in the revised version of our manuscript.**
>
> >Q2: Can you provide additional technical details about the experimental setup. The appendix does not seem to contain any information related to that and only limited information is given in the main paper (what are exactly the training objectives and hyper-parameters such as batch size, etc.).
>
> A2: **In our initial manuscript, the loss of M-MAE is given by equation (11) and the definition of TCR is given by definition 3.4 and hyper-parameters like batch size are given in section 5.1.** We appreciate the reviewer's question and their valuable input. However, based on the information provided, we are unable to fully understand the specific implementation details that are lacking. If possible, we kindly request the reviewer to provide further clarification or specific requirements, so that we can better assist and address their concerns. We are more than happy to provide additional guidance and clarification to ensure that the reviewer receives the necessary and accurate information they require.
>
> >Q3: It is not fully clear how Theorems 1 and 2 are not going against each other given Definition 2. Can you provide some information about that?
>
> A3: **Maximizing mutual information is not contradictory to maximizing joint entropy.** For example, in traditional information theory $X=Y$ $\sim$ uniform distribution, then $H(X)=H(Y)=H(X, Y)=I(X, Y)$ and are all maximized. We have similar cases here in matrix information theory, if $K_1=K_2=I_d$, then $H_2(K_1)=H_2(K_2)=H_2(K_1, K_2)=I_2(K_1, K_2)$ and are all maximized.

---

> ### Author Response · Authors · 2023-11-20
> **We would be grateful if you could take a look at the response**
>
> Dear Reviewer j36B:
>
> We sincerely appreciate your valuable time devoted to reviewing our manuscript. We would like to gently remind you of the approaching deadline for the discussion phase. We have diligently addressed the issues you raised in your feedback, providing detailed explanations. For instance, we have corrected the typo "K1=Z1" in the initial manuscript. We also give an intuitive understanding of why mutual information maximization is not contradictory to joint entropy maximization. Would you kindly take a moment to look at it?
>
> We are very enthusiastic about engaging in more in-depth discussions with you.

---

> > ### Comment · Reviewer_j36B · 2023-11-20
> > **Still not convinced by the theory of the paper**
> >
> > I am still not convinced by the theory of the paper. For instance at the end of Lemma A.1., what does that mean "Thus the above optimization problem *gets a minimum of n*, with x = y = 0 as the unique solution." given that the optimization is over x and y? Also the way this lemma is then used in the proof of Theorem 3.2 is unclear (as also described by reviewer qQv9).

---

> ### Author Response · Authors · 2023-11-21
>
> Thank you for your questions. We will explain them in details as follows.
>
> >Q4: I am still not convinced by the theory of the paper. For instance at the end of Lemma A.1., what does that mean "Thus the above optimization problem gets a minimum of n, with x = y = 0 as the unique solution." given that the optimization is over x and y? Also the way this lemma is then used in the proof of Theorem 3.2 is unclear (as also described by reviewer qQv9).
>
> A4: The minimum here means the minimum objective value is n. "with x = y = 0 as the unique solution." means x=0 and y=0 is the (optimal) point of this optimization problem that gives a minimum (optimal) objective value of n.
>
> To better clarify the confusion you raised, we have changed the statement of our theorems from "Barlow twins and spectral contrastive learning seek the maximal mutual information" into "The optimal point of Barlow twins and spectral contrastive learning losses maximize the matrix mutual information".
>
> We will now explain how this lemma is used in the proof of theorem 3.2.
>
> Take the proof of Barlow twins as an example. Note the diagonal of $K_1$ and $K_2$ are all 1.
>
> Look at the quantity $\frac{\| K_1 \|^2_F \| K_2 \|^2_F}{\| K_1 \odot K_2 \|^2_F}$. From the definition of F norm and the fact that diagonal of $K_1$ and $K_2$ are all 1. We know that this quantity can be rewritten as follows:
> $\frac{(d+ \sum_{i \neq j} (K_1(i,j))^2) (d+ \sum_{i \neq j} (K_2(i,j))^2)}{(d+ \sum_{i \neq j} (K_1(i,j) K_2(i,j))^2)}$.
>
> From Prop 3.1, matrix mutual information is maximized iff $\frac{\| K_1 \|^2_F \| K_2 \|^2_F}{\| K_1 \odot K_2 \|^2_F}$ is minimized.
> Take $(K_1(i,j))^2$ ($i \neq j$) as each elements of x and $(K_2(i,j))^2$ ($i \neq j$) as each elements of y, from the lemma, it is clear that x=y=0 minimizes $\frac{\| K_1 \|^2_F \| K_2 \|^2_F}{\| K_1 \odot K_2 \|^2_F}$ thus maximizes matrix mutual information. Note by construction, $K_1(i,j) = (\bar{z}^{(1)}_i)^T \bar{z}^{(1)}_j$ and $K_2(i,j) = (\bar{z}^{(2)}_i)^T \bar{z}^{(2)}_j$.

---

> ### Author Response · Authors · 2023-11-23
> **Seeking Your Input on Revised Paper's Alignment with ICLR Standards**
>
> Dear Reviewer j36B,
>
> As the discussion period approaches its conclusion, **we want to ensure that we have thoroughly addressed all your concerns and that our revised paper fully meets the standards of ICLR**. We would highly value any additional feedback you may provide.
>
> Thank you sincerely for your time and consideration.
>
> Best regards,
>
> The Authors

---

### Official Review · Reviewer_qQv9 · 2023-10-30

**Soundness:** 2 fair
**Presentation:** 2 fair
**Contribution:** 2 fair
**Rating:** 5
**Confidence:** 4

**Summary:**

This paper shows that the optimal point of BarlowTwins and Spectral Contrastive learning objective functions satisfy the maximal matrix mutual information and the maximal matrix joint entropy. Then, this paper proposes a "matrix variational masked auto-encoder (M-MAE) loss" that is a combination of the original loss and the total coding rate, defined as $\log det (\mu I + ZZ^T)$, where $\mu$ is a hyperparameter. Experimental results show that compared to MAE and U-MAE, the proposed M-MAE performs better in terms of linear probing and fine-tuning when the hyperparameter $\mu$ is well-tuned.

**Strengths:**

This paper attempts to understand self-supervised learning methods in terms of the mutual information maximization framework, where each random variable is from the online encoder and the target encoder. It could be a somewhat valuable attempt to understand how self-supervised learning methods work.

**Weaknesses:**

1. There is no detailed discussion between the introduced matrix entropy (definition 1) and the original Shannon's information entropy. In fact, The Renyi entropy is defined by a special matrix family named density matrix. It is not defined for an arbitrary matrix, but only for a Gram matrix. However, this paper misuses the concept of matrix entropy throughout the whole paper. For example, In page 14, the last paragraph says that "Take K1 = Z1 and K2 = Z2, the results follow similarly". However, K1 and K2 should be Gram matrices, while Z1 and Z2 are feature matrices, that are not a Gram matrix. I think this paper should clarify the relationship between the matrix entropy (defined by a Gram matrix of the samples from a probability distribution) and the original Shannon's information entropy (defined by a random variable following a probability density function).
2. There is no connection between Section 3 and 4. Note that the main difference between MAE and M-MAE is $TCR(Z)$. TCR is used for measuring a joint information quantity in Section 3, but in Section 4, this paper uses TCR for measuring information quantity for a single random variable. As the previous discussions are based on the relationship between Z1 and Z2, the newly introduced regularization term for MAE is irrelevant to the previous results.
3. The proposed M-MAE is sensitive to the choice of the hyperparameter, as shown in Table 2. The gap between each $\mu$ varies a lot, and it means that we need to access the original target labels to tune the hyperparemeter. It violates the spirit of self-supervised learning; we should not access the original labels
4. The results in Section 3 are not generalizable to the generic self-supervised learning methods; these results are only applicable to Barlow twins and spectral contrastive learning. In fact, as mentioned in my previous comment, the proof is wrong for spectral contrastive learning because K1 and K2 should be a Gram matrix. In other words, the proof only works for a special case of self-supervised learning, where the objective function coincides with "Proposition 1" and K1 and K2 are Gram matrices of the feature matrices Z1 and Z2.
5. There is no discussion of why the mutual information maximization (or joint entropy) of Z1, Z2 is a good measure of a good self-supervised learning method. As there is no connection between mutual information maximization and goodness of self-supervised learning methods, the motivation of M-MAE is somewhat weak. What is the benefit of making an MAE model maximize mutual information? (Note that, even more it is actually not about mutual information. See comment 2)
6. The experimental results only show the comparisons between MAE, U-MAE and M-MAE. There are a lot of self-supervised learning methods. I think this paper needs more comparisons with other self-supervised learning methods (e.g., BalowTwins, MoCo, SimCLR, BYOL, DeepClustering, Swav, Data2Vec, DINO, iBot, SimMIM, ...) in terms of both information quantity and performance. I also think that it would be good for this paper to compare with other MAE variants (or MIM methods, such as SimMIM), but it could depend on the scope of this paper; as I think this paper needs a heavy non-trivial revision, as of now, I don't argue that M-MAE should be compared with other MAE variants, but I think additional comparisons with MAE variants will make the submission stronger. I recommend this survey paper to search more recent MAE variants: "A Survey on Masked Autoencoder for Self-supervised Learning in Vision and Beyond"

**Questions:**

Please check my previous comment. I think the current version of this paper will need a non-trivial heavy revision, including re-checking the major motivation (W2, W5), the mathematical notations and theoretical results (W1, W2, W4), adding more experiments (W6), fixing the fundamental flaw -- hyperparameter sensitivity -- of the proposed method (W3).

---

> ### Author Response · Authors · 2023-11-19
> **Total response to reviewer qQv9**
>
> Thank you for taking your time to review our paper. As your concerns (W1, W2, W4, W5) are caused by our typo. We will make further clarification here. As our matrix information quantities are defined on **square** matrices whose diagonals are all $1$. Taking $K_1=Z_1$ and $K_2=Z_2$ is a typo, the correct one should be $K_1=Z^T_1Z_1$ and $K_2=Z^T_2Z_2$. **This is why we say in the proof part that it is similar to Barlow twins loss proof where the $K_1$ and $K_2$ defined there are square matrices whose diagonals are all $1$.** We have corrected this typo in the revised manuscript.
>
> We propose to use a unified matrix information-theoretic view on self-supervised learning. We want to further stress that our main contributions are given a theoretical understanding of both contrastive and non-contrastive methods and we also use matrix information theory to improve masked image modelling.

---

> ### Author Response · Authors · 2023-11-19
> **Response 1 to reviewer qQv9**
>
> >Q1: There is no detailed discussion between the introduced matrix entropy (definition 1) and the original Shannon's information entropy. In fact, The Renyi entropy is defined by a special matrix family named density matrix. It is not defined for an arbitrary matrix, but only for a Gram matrix. However, this paper misuses the concept of matrix entropy throughout the whole paper. For example, In page 14, the last paragraph says that "Take K1 = Z1 and K2 = Z2, the results follow similarly". However, K1 and K2 should be Gram matrices, while Z1 and Z2 are feature matrices, that are not a Gram matrix. I think this paper should clarify the relationship between the matrix entropy (defined by a Gram matrix of the samples from a probability distribution) and the original Shannon's information entropy (defined by a random variable following a probability density function).
>
> A1: First of all, we want to point out that neither Renyi (matrix) entropy nor Shannon's information entropy are defined by us, we merely use the standard definition in the literature. Secondly, although both are entropies, these two types of entropy are defined from completely different perspectives. This means that it is not straightforward to simply transfer the self-supervised results based on Shannon entropy to Renyi (matrix) entropy with slight modifications. Therefore, we need to start from scratch, and this is not a trivial outcome. We also discuss why we use matrix information theory instead of traditional information theory in the first few paragraphs in section 3.1 of the initial manuscript.
>
> **For the "Take K1=Z1" typo, we have discussed this in the general response and total response, we have corrected this typo.** The relationship between Renyi (matrix) entropy and Shannon entropy is that Renyi entropy can be seen as applying Shannon entropy to the spectrum of (density) matrix. We have already discussed this in the Related work part of our initial manuscript. We want to further emphasize that in this paper, our focus are mainly on **apply** the tools from matrix information theory.
>
> >Q2: There is no connection between Section 3 and 4. Note that the main difference between MAE and M-MAE is TCR(Z). TCR is used for measuring a joint information quantity in Section 3, but in Section 4, this paper uses TCR for measuring information quantity for a single random variable. As the previous discussions are based on the relationship between Z1 and Z2, the newly introduced regularization term for MAE is irrelevant to the previous results.
>
> A2: Section 3 deals with contrastive and feature-decorrelation based SSL methods. As we point out, in traditional information theory, when the Siamese structure degenerates into a single branch like MAE, the joint entropy and mutual information are exactly the entropy. So motivated by section 3, we add a entropy regularizer TCR. **We have already discussed this connection of section 3 and 4 at the beginning of section 4 in our initial manuscript.** Please note that our TCR is performed on matrices, and it can be computed either jointly or individually, depending on the input matrix. On the other hand, in Section 4, we focus on computation based on representations. We have never claimed that there is a theoretical result on MAE, the theoretical results from the Siamese architecture only served as motivation for improvements in the MAE approach (only one branch).
>
> >Q3: The proposed M-MAE is sensitive to the choice of the hyperparameter $\mu$, as shown in Table 2. The gap between each
>  varies a lot, and it means that we need to access the original target labels to tune the hyperparemeter. It violates the spirit of self-supervised learning; we should not access the original labels
>
> A3: Robustness may not be our primary focus; rather, we are highlighting the potential for improvement with matrix information theory. Additionally, this theoretically motivated improvement is closely connected to previous method U-MAE. **Moreover, the closely related method U-MAE is also sensitive to hyper-parameter.**

---

> ### Author Response · Authors · 2023-11-19
> **Response 2 to reviewer qQv9**
>
> >Q4: The results in Section 3 are not generalizable to the generic self-supervised learning methods; these results are only applicable to Barlow twins and spectral contrastive learning. In fact, as mentioned in my previous comment, the proof is wrong for spectral contrastive learning because K1 and K2 should be a Gram matrix. In other words, the proof only works for a special case of self-supervised learning, where the objective function coincides with "Proposition 1" and K1 and K2 are Gram matrices of the feature matrices Z1 and Z2.
>
> A4: **As we have discussed above, the typo of $K_1=Z_1$ may make you confused, we have corrected this typo.** Secondly, using the concept introduced in the paper [1], spectral contrastive learning is a special type of sample-contrastive method and Barlow twins is a special type of dimension-contrastive method. **Following the exactly same proof of our theorem, our theoretical results can be generalized to sample contrastive and dimension contrastive methods.** As pointed out by [1], sample and dimension contrastive methods contain many generic self-supervised methods (Proposition 3.2 of [1]).
>
> [1]: On the duality between contrastive and non-contrastive self-supervised learning, Quentin Garrido, Yubei Chen, Adrien Bardes, Laurent Najman, Yann LeCun, ICLR 2023 (notable-top-5%, ICLR 2023 Outstanding Paper Honorable Mention)
>
> >Q5: There is no discussion of why the mutual information maximization (or joint entropy) of Z1, Z2 is a good measure of a good self-supervised learning method. As there is no connection between mutual information maximization and goodness of self-supervised learning methods, the motivation of M-MAE is somewhat weak. What is the benefit of making an MAE model maximize mutual information? (Note that, even more it is actually not about mutual information. See comment 2)
>
> A5: Our theoretical results show that the loss function of many SSL methods can be seen as **exactly** maximizing matrix information theoretic quantities (matrix mutual information or matrix entropy). As the algorithms we analyze are empirically good and our theorems show **exact** maximization relationship of matrix information quantities and "good" SSL methods' losses. We think this connection is straight forward. For why adding matrix entropy may improve MAE, as we pointed out the **exact** relationship of effective rank and matrix entropy, [2] shows in their proposition 5.1 that bigger effective rank is better for SSL. **Note these motivations are already discussed in our initial paper.**
>
> [2]: RankMe: Assessing the Downstream Performance of Pretrained Self-Supervised Representations by Their Rank, Quentin Garrido, Randall Balestriero, Laurent Najman, Yann LeCun, ICML 2023
>
> >Q6: The experimental results only show the comparisons between MAE, U-MAE and M-MAE. There are a lot of self-supervised learning methods. I think this paper needs more comparisons with other self-supervised learning methods (e.g., BalowTwins, MoCo, SimCLR, BYOL, DeepClustering, Swav, Data2Vec, DINO, iBot, SimMIM, ...) in terms of both information quantity and performance. I also think that it would be good for this paper to compare with other MAE variants (or MIM methods, such as SimMIM), but it could depend on the scope of this paper; as I think this paper needs a heavy non-trivial revision, as of now, I don't argue that M-MAE should be compared with other MAE variants, but I think additional comparisons with MAE variants will make the submission stronger. I recommend this survey paper to search more recent MAE variants: "A Survey on Masked Autoencoder for Self-supervised Learning in Vision and Beyond"
>
> A6: There may be unfair comparisons when comparing our method with other SSL approaches. The MAE and its variants have different underlying principles compared to other self-supervised methods. Additionally, U-MAE only compares with MAE variants. Note we plan to add a reproduction of SimMIM in the future. We agree with you that "I don't argue that M-MAE should be compared with other MAE variants" and our comparisons already cover the most related and canonical methods MAE and U-MAE. **It's worth noting that we have cited the paper "A Survey on Masked Autoencoder for Self-supervised Learning in Vision and Beyond" to provide a broader context for our work.**

---

> ### Author Response · Authors · 2023-11-19
> **Response 3 to reviewer qQv9**
>
> >Q7: Please check my previous comment. I think the current version of this paper will need a non-trivial heavy revision, including re-checking the major motivation (W2, W5), the mathematical notations and theoretical results (W1, W2, W4), adding more experiments (W6), fixing the fundamental flaw -- hyperparameter sensitivity -- of the proposed method (W3).
>
> A7: Thank you for your feedback. **We have made minor revisions, which consist of correcting the typos and add a new reference.**
>
> We also conduct additional experiments on CIFAR-100 (pretrained for 1k epochs), the linear probing and fine-tuning accuracies are as follows:
>
> | Method | linear@1 | linear@5 | finetune@1 | finetune@5 |
> | --- | --- | --- | --- | --- |
> | M-MAE(vit-base) | **60.9** | **88.1** | 83.8 | **97.0** |
> | U-MAE(vit-base) | 59.7 | 86.8 | 84.1 | 96.5 |
> | MAE(vit-base) | 59.2 | 86.8 | 84.5 | 96.6 |
>
>
> It is clear that our method has good results compared to existing works, which is promising as we don't have time to search for better hyper-parameters during rebuttal period.

---

> > ### Comment · Reviewer_qQv9 · 2023-11-20
> >
> > I have tried to understand the proof of Theorem 3.2. As the comment by Reviewer j36B, the "proof" looks incomplete. As far as I understood, the proof argues that
> >
> > 1. Mutual information is maximized when the RHS of proposition 3.1 is minimized.
> > 2. The RHS of Prop3.1 has a solution when each element of $K\_1$ and $K\_2$ is zero (i.e., when $\| K_1 \|_F^2 = \| K_2 \|_F^2 = 0$)
> > 3. Barlow Twins **"encourages"** $z_i^{(1)} \approx z_i^{(2)}$ and $z_i^{(1)} \approx z_j^{(2)} \approx 0$
> > 4. We can **approx** $z_i^{(1)} \approx z_i^{(2)} \approx 0$
> > 5. The RHS of Prop3.1 becomes zero when $z_i^{(1)} \approx z_i^{(2)} \approx 0$
> > 6. Therefore, Balow Twins is the same as MI maximization
> > 7. Assume $K_i Z_i^T Z_i$ and repeat 3-6, then spectral contrastive learning is also MI maximisation.
> >
> > This looks like incomplete (and actually technically unacceptable) proof in terms of math because,
> >
> > - (The statement and the proof solve a different problem) When we say "a loss function is the **exactly** same as another loss function", we should show that the two functions are reduced in the same form, not a solution having a loss value 0 for function B also satisfies making function A's value 0. It is because, in practice, function B will rarely become 0. In other words, it could be impossible to make $\| K_1 \|_F^2 = \| K_2 \|_F^2 = 0$ by function B (SSL losses) therefore, the intermediate solution of function B will not minimize function A (matrix MI, in this case). If we do not have latent embeddings satisfying the approximation, the proof will not hold anymore. This should be clarified in the proof.
> > - (Number 3-6 is an improper **proof**) How can we define "encourages"? Will Barlow Twins always make $z_i^{(1)} \approx z_i^{(2)}$ and $z_i^{(1)} \approx z_j^{(2)} \approx 0$? Number 3-6 shows that just assuming two convenient **orthogonal** feature matrices, then the matrix MI will be maximized. However, it cannot guarantee that Barlow Twins, Spectral contrastive learning loss is the **exactly** same as the mutual information maximisation.
> >
> > Sometimes, we may accept this kind of non-rigorous proof, if the paper has more advantages (and if the revised paper tones down the overall argument). However, I found this paper has other flaws as well.
> >
> > [A1] Still, there is no specific discussion regarding matrix entropy. What is matrix entropy? What does it mean? How is it related to classical information entropy? I feel that we need a more precise understanding of matrix entropy before just applying them as a tool.
> >
> > [A2] "the joint entropy and mutual information are exactly the entropy". Joint entropy and mutual information are defined for two random variables, while entropy is defined for a single random variable. I think it is a misuse of the terminologies joint entropy, mutual information and information entropy. I don't think the knowledge of joint entropy, mutual information is directly applied to the single random variable entropy. It is not trivial.
> >
> > [A3] I don't agree the argument. This paper tackles "self-supervised learning", which aims to learn representations without accessing labels. However, the method hacks the SSL benchmark by directly tuning the hyperparameter on the test set. I don't think it is a reliable evaluation. The sensitivity of U-MAE does not resolve this weakness. It does means that U-MAE also has the same weakness.
> >
> > [A4] Although it is clarified, I think the proof is somewhat nonrigorous, as my first comment. What if a loss function cannot make $z_1 = z_2$? Certainly, a teacher-student framework (or online-target framework) will aim to make $z_1 = z_2$, but it is hard to say that such framework actually ensures to make $z_1 = z_2$.
> >
> > [A5] As my previous comment, if one says **A is exactly the same as B", the proof should be more rigorous. Furthermore, even if we can argue that SSL is exactly the same as matrix information maximisation, it does not support that it is a good measure of a good self-supervised learning method. I still feel that there is no connection between mutual information maximization and the goodness of self-supervised learning methods
> >
> > [A6] "The MAE and its variants have different underlying principles compared to other self-supervised methods" what is the different underlying principle of MAE? It has the same SSL spirit as the others. We should not have access to the target labels. MAE has a certain difference from others: MAE is evaluated by fine-tuning rather than linear probing, but I don't think it makes a different principle from others.
> >
> >
> > I raised some concerns in my initial review. I feel that most of them are not well addressed in the rebuttal. Particularly, I think this paper needs more rigorous proof than the current version (as far as I understood, Reviewer j36B has a similar concern to me). Also, I think the experiment violates the spirit of SSL (no access to target labels). Overall, I think this paper needs more improvements to be accepted at ICLR.

---

> > > ### Comment · Reviewer_qQv9 · 2023-11-20
> > >
> > > **Additional comment regarding the proof for clarification**
> > >
> > > Eventually, **Theorem 3.2** shows that the matrix mutual information between two features is maximized when two features are the same, where it is a fairly acceptable argument even without proof. I don't argue that the proof for showing this is wrong (i.e., before "As the loss of Barlow Twins ...").
> > >
> > > However, I don't think it is proof for the statement *"Barlow twins and spectral contrastive learning seek the maximal mutual information"*, because there is no guarantee that Barlow twins and spectral contrastive learning guarantee that the two features (target and online features) become the same.

---

> ### Author Response · Authors · 2023-11-20
> **We would be grateful if you could take a look at the response**
>
> Dear Reviewer qQv9:
>
> We sincerely appreciate your valuable time devoted to reviewing our manuscript. We would like to gently remind you of the approaching deadline for the discussion phase. We have diligently addressed the issues you raised in your feedback, providing detailed explanations. For instance, we have corrected the typo "K1=Z1" in the initial manuscript, which may cause confusion when you first read the initial manuscript. Would you kindly take a moment to look at it?
>
> We are very enthusiastic about engaging in more in-depth discussions with you.

---

> ### Author Response · Authors · 2023-11-21
> **Response 1 to reviewer qQv9's new questions**
>
> Thank you for taking the time to review our paper and considering our rebuttal.
> **However, it appears that the reviewer may have some misunderstandings about certain aspects of our paper, particularly our proof of Theorem 3.2.** In order to provide further clarification and address the reviewer's concerns more comprehensively, we will address each of the reviewer's questions individually as follows.
>
>
> >Q8: I have tried to understand the proof of Theorem 3.2. As far as I understood, the proof argues that ... (7 steps).
>
> A8: We'd like to make inline comments to the 7 steps provided by the reviewer.
>
> >1. Mutual information is maximized when the RHS of proposition 3.1 is minimized.
>
> A: Your understanding of this point is correct.
>
> >2. The RHS of Prop3.1 has a solution when each element of $K_1$ and $K_2$ is zero (i.e., when $\|K_1\|^2_F = \|K_1\|^2_F=0$)
>
> A: Your understanding of this point is **incorrect**. The minimum will be attained if each **off-diagonal** element in $K_1$ and $K_2$ is 0. Also, the F norm will **not** be 0, because each diagonal element in $K_1$ and $K_2$ will be 1.
>
> >3. Barlow Twins "**encourages**" $z^{(1)}_i \approx z^{(2)}_i$ and $z^{(1)}_i \approx z^{(2)}_j \approx 0$
>
> A: Your understanding of this point is **incorrect**. Minimizing the loss will encourage $(z^{(1)}_i)^T z^{(2)}_j \approx 0$ not $z^{(1)}_i \approx z^{(2)}_j \approx 0$.
>
> >4. We can **approx** $z^{(1)}_i \approx z^{(2)}_i \approx 0$
>
> A: Your understanding of this point is **incorrect**. You can only obtain $z^{(1)}_i \approx z^{(2)}_i$.
>
> >5. The RHS of Prop3.1 becomes zero when $z^{(1)}_i \approx z^{(2)}_j \approx 0$
>
> A: Your understanding of this point is **incorrect**. The RHS of Prop 3.1 is minimized when $z^{(1)}_i = z^{(2)}_i$ and $(z^{(1)}_i)^Tz^{(2)}_j=0$.
>
> >6. Therefore, Balow Twins is the same as MI maximization
>
> A: Your understanding of this point is correct.
>
> >7. Assume $K_iZ^T_iZ_i$ and repeat 3-6, then spectral contrastive learning is also MI maximisation.
>
> A: Your understanding of this point is correct.
>
>
> >Q9: This looks like incomplete (and actually technically unacceptable) proof in terms of math because, ......If we do not have latent embeddings satisfying the approximation, the proof will not hold anymore. This should be clarified in the proof.
>
> A9: **We did not say "a loss function is the exactly same as another loss function".** Instead, we show that when these loss functions reach their optimal points the matrix mutual information and matrix joint entropy are maximized. Secondly, we have consistently emphasized that our analysis focuses on the loss functions themselves. In practice, factors such as optimization algorithms and network expressive power can affect the ability to optimize towards optimal points but these are out of the scope of this paper. Analyzing the loss directly is a common approach, as seen, for example, in the work of [1].
>
> [1]: On the duality between contrastive and non-contrastive self-supervised learning, Quentin Garrido, Yubei Chen, Adrien Bardes, Laurent Najman, Yann LeCun, ICLR 2023 (notable-top-5%, ICLR 2023 Outstanding Paper Honorable Mention)
>
> >Q10: (Number 3-6 is an improper proof) How can......same as the mutual information maximisation.
>
> A10: We use **"encourage"** here following the spirit of prior work "Understanding Contrastive Representation Learning through Alignment and Uniformity on the Hypersphere". As what you summarized in your summary part of our paper "This paper shows that the optimal point of BarlowTwins and Spectral Contrastive learning objective functions satisfy the maximal matrix mutual information and the maximal matrix joint entropy". The "encourage" here means the optimal point behavior of losses. To better clarify the confusion you raised, we have changed the statement of our theorems from "Barlow twins and spectral contrastive learning seek the maximal mutual information" into "The optimal point of Barlow twins and spectral contrastive learning losses maximize the matrix mutual information". The **approx**  in the proof will also become equality, other stuff in the proof will remain unchanged.
>
>
> Understanding Contrastive Representation Learning through Alignment and Uniformity on the Hypersphere (https://arxiv.org/pdf/2005.10242.pdf), citation>1200.
>
> The following **taken from section 4 (named Feature Distribution on the Hypersphere ) of their paper**
>
> "The contrastive loss **encourages** learned feature representation for positive pairs to be similar, while pushing features from the randomly sampled negative pairs apart."

---

> ### Author Response · Authors · 2023-11-21
> **Response 2 to reviewer qQv9's new questions**
>
> >Q11: [A1] Still, there is no specific discussion regarding matrix entropy. What is matrix entropy? What does it mean? How is it related to classical information entropy? I feel that we need a more precise understanding of matrix entropy before just applying them as a tool.
>
> A11: **In our previous rebuttal, we explicitly pointed out that the discussion on relationship is included in the related work part, can be understood in terms of spectrum.** For a detailed discussion, we refer you to the paper we cited (DiME: Maximizing Mutual Information by a Difference of Matrix-Based Entropies, https://arxiv.org/pdf/2301.08164.pdf), where the background section provides a thorough explanation. As we pointed out previously, we did not extensively discuss matrix entropy because it is not our focal point and not a concept we defined.
>
>
> >Q12: [A2] "the joint entropy and mutual information are exactly the entropy". Joint entropy and mutual information are defined for two random variables, while entropy is defined for a single random variable. I think it is a misuse of the terminologies joint entropy, mutual information and information entropy. I don't think the knowledge of joint entropy, mutual information is directly applied to the single random variable entropy. It is not trivial.
>
> A12: **Please provide a complete restatement of our original rebuttal without only quoting a portion.** The initial response we provided is "As we point out, in traditional information theory, when the Siamese structure degenerates into a single branch like MAE, the joint entropy and mutual information are exactly the entropy." not just the last half you mentioned. In traditional information theory, when there are two random variables X=Y, then the joint entropy H(X, Y)= mutual information I(X;X) = H(X) = H(Y).
>
> >Q13: [A3] I don't agree the argument. This paper tackles "self-supervised learning", which aims to learn representations without accessing labels. However, the method hacks the SSL benchmark by directly tuning the hyperparameter on the test set. I don't think it is a reliable evaluation. The sensitivity of U-MAE does not resolve this weakness. It does means that U-MAE also has the same weakness.
>
> A13: We agree with your statement. As U-MAE can be seen as our second-order approach, it is natural that our "weaknesses" align with those of U-MAE. It is worth noting that U-MAE is a pioneering work that modifies the MAE loss based on theoretical foundations, and we are pleased to have found close connections to U-MAE. **Additionally, we would like to emphasize that sensitivity to hyperparameters in SSL is quite normal (as seen in renowned works like SimCLR and VICReg, which require searching for suitable hyperparameters).**
>
> SimCLR (https://arxiv.org/pdf/2002.05709.pdf),Table 5 in Section 5.1, citation> 13000
>
> VICReg (https://arxiv.org/pdf/2105.04906.pdf),Table 7 in Section D.4, citation > 600
>
> >Q14: [A4] Although it is clarified, I think the proof is somewhat nonrigorous, as my first comment. What if a loss function cannot make $z_1=z_2$? Certainly, a teacher-student framework (or online-target framework) will aim to make $z_1=z_2$, but it is hard to say that such framework actually ensures to make $z_1=z_2$.
>
> A14: **We have consistently emphasized that our analysis focuses on the loss functions themselves.** In practice, factors such as optimization algorithms and network expressive power can affect the ability to optimize towards optimal points but these are out of the scope of this paper.  To better clarify the confusion you raised, we have changed the statement of our theorems from "Barlow twins and spectral contrastive learning seek the maximal mutual information" into "The optimal point of Barlow twins and spectral contrastive learning losses maximize the matrix mutual information". The **approx**  in the proof will also become equality, other stuff in the proof will remain unchanged.

---

> ### Author Response · Authors · 2023-11-21
> **Response 3 to reviewer qQv9's new questions**
>
> >Q15: [A5] As my previous comment, if one says **A is exactly the same as B", the proof should be more rigorous. Furthermore, even if we can argue that SSL is exactly the same as matrix information maximisation, it does not support that it is a good measure of a good self-supervised learning method. I still feel that there is no connection between mutual information maximization and the goodness of self-supervised learning methods
>
> A15:  **Firstly, from a theoretical perspective.** Following the exactly same proof of our theorem, our theoretical results can be generalized to sample contrastive and dimension contrastive methods. As pointed out by [1], sample and dimension contrastive methods contain many "good" self-supervised methods (Proposition 3.2 of [1]). **Secondly, from an empirical perspective,** our matrix information quantities act as strong indicator for good representation. From the matrix mutual information plot is in the Figure 1 (https://i.postimg.cc/J7bhbTdm/matrix-mutual-information.png) and the matrix joint entropy plot is in Figure 2 (https://i.postimg.cc/Pr0yD5LP/matrix-joint-entropy.png). We set temperatures as 0.3, 0.5, 0.7 in SimCLR to plot these Figures. From Figure 1 and 2, we can observe that the increase of matrix mutual information or matrix joint entropy during training ties closely with the final accuracy. As temperature = 0.3 outperforms 0.5 and 0.7 in KNN accuracy, it also has the biggest matrix mutual information and matrix joint entropy value.
>
>
> [1]: On the duality between contrastive and non-contrastive self-supervised learning, Quentin Garrido, Yubei Chen, Adrien Bardes, Laurent Najman, Yann LeCun, ICLR 2023 (notable-top-5%, ICLR 2023 Outstanding Paper Honorable Mention)
>
> >Q16: [A6] "The MAE and its variants have different underlying principles compared to other self-supervised methods" what is the different underlying principle of MAE? It has the same SSL spirit as the others. We should not have access to the target labels. MAE has a certain difference from others: MAE is evaluated by fine-tuning rather than linear probing, but I don't think it makes a different principle from others.
>
> A16: **We understand your viewpoint, and we gracefully disagree. We think MAE is quite different from the other SSL methods.**
>
>
>
> >Q17: Additional comment regarding the proof for clarification
> Eventually, Theorem 3.2 shows that the matrix mutual information between two features is maximized when two features are the same, where it is a fairly acceptable argument even without proof. I don't argue that the proof for showing this is wrong (i.e., before "As the loss of Barlow Twins ...").
> However, I don't think it is proof for the statement "Barlow twins and spectral contrastive learning seek the maximal mutual information", because there is no guarantee that Barlow twins and spectral contrastive learning guarantee that the two features (target and online features) become the same.
>
> A17:  **We also do not agree with your summary** of  "Eventually, Theorem 3.2 shows that the matrix mutual information between two features is maximized when two features are the same, where it is a fairly acceptable argument even without proof.". In fact, we show matrix mutual information maximized when $z^{(1)}_i = z^{(2)}_i $ and $(z^{(1)}_i)^T(z^{(2)}_j)=0$ in the proof of Theorem 3.2. **We have consistently emphasized that our analysis focuses on the loss functions themselves.** In practice, factors such as optimization algorithms can affect the ability to optimize towards optimal points but these are out of the scope of this paper. Analyzing the loss directly is a common approach, for example, in the work of [1]. To better clarify the confusion you raised, we have changed the statement of our theorems from "Barlow twins and spectral contrastive learning seek the maximal mutual information" into "The optimal point of Barlow twins and spectral contrastive learning losses maximize the matrix mutual information".
>
> [1]: On the duality between contrastive and non-contrastive self-supervised learning, Quentin Garrido, Yubei Chen, Adrien Bardes, Laurent Najman, Yann LeCun, ICLR 2023 (notable-top-5%, ICLR 2023 Outstanding Paper Honorable Mention)

---

> ### Comment · Reviewer_qQv9 · 2023-11-21
>
> Sorry for my typos at the beginning of the response (e.g., $z \approx 0$). I made mistakes when I wrote the comments from my notes (it could be slightly out of context, but the current notations are really really confusing. I would like to encourage the authors to make the notations clearer, if possible). The inline comments by the authors are indeed correct.
>
> [A9] The rebuttal specifies "a loss function is the **exactly** same as another loss function", e.g., "Our theoretical results show that the loss function of many SSL methods can be seen as **exactly** maximizing matrix information theoretic quantities". Please check [A10] for the overall comment of the theoretical part.
>
> > A5: Our theoretical results show that the loss function of many SSL methods can be seen as exactly maximizing matrix information theoretic quantities (matrix mutual information or matrix entropy). As the algorithms we analyze are empirically good and our theorems show exact maximization relationship of matrix information quantities and "good" SSL methods' losses. We think this connection is straight forward. For why adding matrix entropy may improve MAE, as we pointed out the exact relationship of effective rank and matrix entropy, [2] shows in their proposition 5.1 that bigger effective rank is better for SSL. Note these motivations are already discussed in our initial paper.
>
> [A10] "The approx in the proof will also become equality, other stuff in the proof will remain unchanged". I mean that the assumption makes the approximation (or equality, as the revised version) itself is not rigorous. As my previous comment, the proof fundamentally shows that "the matrix mutual information between two features is maximized when two features are the same". I don't think this statement is extendable to the current format. For example, a classical approach to show whether the optimal point is the same is whether two methods have the same Lagrange multiplier. I feel that assuming method A makes $z^1 = z^2$ is a logical jump.
>
> [A11] "As we pointed out previously, we did not extensively discuss matrix entropy because it is not our focal point and not a concept we defined." As this paper mainly employs matrix entropy, I think this paper should be built upon a high understanding of matrix entropy. As my previous comment,  I feel that we need a more precise understanding of matrix entropy before just applying them as a tool ([A1])
>
> [A12] The current answer makes sense. I thought that the original response ("when the Siamese structure degenerates into a single branch like MAE") does not mean to set H(X, X), but I thought that it introduces a new single variable entropy again.
>
> [A13] In my opinion, the fully fine-tuning evaluation as MAE protocol and the linear probing evaluation as SimCLR and VICReg are not directly comparable in terms of the test set hacking. However, I think it could be somewhat acceptable if the other reviewers and the area chair agree that the hyperparameter sensitivity and the test set hacking are not a serious problem as reject.
>
> [A14] Please check [A10].
>
> [A15] I think the answer is not a perfect one, but it could be an alternative with an empirical analysis. I think citing the proposition of Garrido et al. could be helpful for making the paper better
>
> [A16] "MAE is quite different from the other SSL methods" does not mean that it is okay to hack the test set for SSL. As my comment in [A13], I think the current evaluation needs an agreement among all the reviewers and AC whether the evaluation protocol has no problem.
>
> [A17] Thanks for the clarification. I forgot to mention the second condition. I will re-state my argument: Theorem 3.2 shows that the matrix mutual information between two features is maximized when two Gram matrices are the same and each feature dimension is invariant -- the Gram matrix has 0 off-diagonal.
>
> Basically, I think the paper should be revised as soon as possible. I made my decision based on the initial version, and the current revised paper does not show significant difference with the initial one.
>
> I think some of my concerns are addressed in the response, and some are not. I think the current theoretical analysis is not acceptable (even if the typo is corrected). If the paper argues for a weaker theoretical contribution with proper citations and additional logic, the weakness becomes weaker than the initial version. I think the presentation of the current paper should be revised, including many clarifications in the responses. Finally, I still think that this paper violates the spirit of SSL by hacking the test set with direct hyperparameter tuning, but I will respect all the opinions of the other reviewers and AC.
>
> Please update the paper. I will re-evaluate the paper based on the revised version.

---

> > ### Comment · Reviewer_qQv9 · 2023-11-21
> >
> > Sorry for another mistake in my previous comment. I wrote that the paper is not revised from the initial version based on https://openreview.net/revisions?id=WfjJOEfAf7. I just realised that I actually checked the revised paper. I think the `revision` is only visible to the authors. I updated my comment as "I made my decision based on the initial version, and the current revised paper does not show significant difference with the initial one" to avoid another misunderstanding between us.
> >
> > This message is a private message to authors (as well as chairs), to avoid messy public comments

---

> ### Author Response · Authors · 2023-11-22
> **Response 4 to reviewer qQv9's new questions**
>
> >Q18: This is a question to our response [A9]. The rebuttal specifies "a loss function is the exactly same as another loss function"......Note these motivations are already discussed in our initial paper.
>
> A18: As our initial response to [A9]. The wrong thing in the sentence you raised "a loss function is the exactly same as another loss function" is not the word **exactly**. We highlight the wrong place as "a loss function is the exactly same as **another loss function**". In fact, the matrix information quantities are not a **loss function** in Theorems 3.2 and 3.8. What we prove is that optimal point of losses will make matrix information **quantities** also achieve its optimal.
>
>
>
> >Q19: This is a question to our response [A10]. "The approx in the proof will also become equality, other stuff in the proof will remain unchanged". I mean that the assumption makes the approximation (or equality, as the revised version) itself is not rigorous. As my previous comment, the proof fundamentally shows that "the matrix mutual information between two features is maximized when two features are the same". I don't think this statement is extendable to the current format. For example, a classical approach to show whether the optimal point is the same is whether two methods have the same Lagrange multiplier. I feel that assuming method A makes $z^1=z^2$ is a logical jump.
>
> A19: Your statement "As my previous comment, the proof fundamentally shows that "the matrix mutual information between two features is maximized when two features are the same"." is wrong. **Yourselves find it wrong in response to our answer [A17].** From your next question, take the loss $\|z^1-z^2\|^2$ for example, its optimal is obtained when $z^1=z^2$. Thus, we do not agree with the point you raised "For example, a classical approach to show whether the optimal point is the same is whether two methods have the same Lagrange multiplier.".
>
>
> >Q20: This is a question to our response [A11]. "As we pointed out previously, we did not extensively discuss matrix entropy because it is not our focal point and not a concept we defined."......as a tool ([A1])
>
> A20: As we emphasized previously, this paper mainly uses matrix information theory as a tool. **All the matrix information theoretic quantities are well-defined and easy to understand from their definition.**
>
>
>
> >Q21: This is a question to our response [A12]. The current answer makes sense. I thought that the original response ("when the Siamese structure degenerates into a single branch like MAE") does not mean to set H(X, X), but I thought that it introduces a new single variable entropy again.
>
> A21: **We are glad that you found that we are correct and you misunderstood what we discussed previously.**
>
>
>
> >Q22: This is a question to our response [A13]. In my opinion, the fully fine-tuning evaluation as MAE protocol and the linear probing evaluation as SimCLR and VICReg are not directly comparable in terms of the test set hacking. However, I think it could be somewhat acceptable if the other reviewers and the area chair agree that the hyperparameter sensitivity and the test set hacking are not a serious problem as reject.
>
> A22: **We do ablation studies on hyper-parameters just like the work SimCLR and VICReg did.** So we can not agree with you that "In my opinion, the **fully fine-tuning** evaluation as MAE protocol and the linear probing evaluation as SimCLR and VICReg are not directly comparable in terms of the test set hacking." **Note the ablation study we done is linear probing.**
>
>
>
> >Q23: This is a question to our response [A14]. Please check [A10].
>
> A23: As your question is same as Q19. We have already discussed it.
>
>
> >Q24: This is a question to our response [A15]. I think the answer is not a perfect one, but it could be an alternative with an empirical analysis. I think citing the proposition of Garrido et al. could be helpful for making the paper better
>
> A24: Thank you for your suggestion. We will add a new remark at the end of Appendix A based on our answer [A15].

---

> ### Author Response · Authors · 2023-11-22
> **Response 5 to reviewer qQv9's new questions**
>
> >Q25: This is a question to our response [A16]. "MAE is quite different from the other SSL methods" does not mean that it is okay to hack the test set for SSL. As my comment in [A13], I think the current evaluation needs an agreement among all the reviewers and AC whether the evaluation protocol has no problem.
>
> A25: We have discussed in our initial response [A16]. We do ablation studies on hyper-parameters just like the work SimCLR and VICReg did. Moreover, in the initial MAE paper (https://arxiv.org/pdf/2111.06377.pdf), its Figure 5 is ablation study on the hyper-parameter mask ratio. **In MAE's Figure 5, we can see its linear probing accuracy drops from 71.8 to 66.1 when mask ratio changes from 0.8 to 0.9.**
>
> >Q26: This is a question to our response [A17]. Thanks for the clarification. I forgot to mention the second condition. I will re-state my argument: Theorem 3.2 shows that the matrix mutual information between two features is maximized when two Gram matrices are the same and each feature dimension is invariant -- the Gram matrix has 0 off-diagonal.
>
> A26: **We are glad that you found that we are correct and you misunderstood what we discussed previously.**

---

> ### Author Response · Authors · 2023-11-23
> **Seeking Your Input on Revised Paper's Alignment with ICLR Standards**
>
> Dear Reviewer qQv9,
>
> As the discussion period approaches its conclusion, **we want to ensure that we have thoroughly addressed all your concerns and that our revised paper fully meets the standards of ICLR**. We would highly value any additional feedback you may provide.
>
> Thank you sincerely for your time and consideration.
>
> Best regards,
>
> The Authors

---

> ### Comment · Reviewer_qQv9 · 2023-11-23
>
> Dear authors,
>
> Before starting, I listed my initial concerns:
>
> - Re-checking the major motivation (W2, W5)
> - The mathematical notations and theoretical results (W1, W2, W4)
> - Insufficient experiments (W6)
> - Fixing the fundamental flaw -- hyperparameter sensitivity -- of the proposed method (W3)
>
> The rationale for the strong reject was mainly because (1) I suspect that the concept of matrix entropy is misused without a careful understanding of the concept, (2) I think the empirical contribution is also somewhat insufficient. Therefore, I evaluated this paper as having poor soundness and contribution. Similarly, as the link between Section 3 and 4 is not clear and the concept of the matrix information is not properly provided, I evaluated the presentation as 1.
>
> After long discussions, I re-evaluated the paper based on the revised version as weak reject (5).
>
> - First, my strongest concern was the concept of matrix entropy itself is misused by the paper due to the "typo" in the paper. I don't think the current theoretical results are precise enough (e.g., "encouraging" or "assuming the solutions" is not proper proof in my opinion), but I think some of the theoretical contributions are acceptable but below the borderline. Regarding **Misinterpretation of simple math concepts** in the comment for Reviewer znaA, I clarified that I did make a mistake when I wrote down notations in markdown, not did not mean that I misunderstood the theorem. I correctly understood the proof and therefore raised concerns.
> - Second, I still think that Section 3 and 4 are somewhat unrelated. Section 3 is for contrastive learning methods and Section 4 is newly introduced for MAE. Therefore I raised concerns about (1) Section 3 is based on joint entropy but 4 is based on single entropy. Although the authors clarified that the joint entropy of self-variables is the same as the single entropy, but I still think that the connection between theorems in section 3 and the newly introduced regularization in section 4 are not well linked. I partially agree with the comments (e.g., by citing DiME), but these discussions are not included in the revised paper. Even if assuming all the comments are updated in the revised paper, I am still slightly skeptical about the relationship between the theorems and the proposed regularization method
> - Third, in terms of the evaluation, I still think that directly searching for the best hyperparameter on the test set is not acceptable. However, (1) assuming that the main contribution of this paper is based on bridging the concept of matrix information and SSL (although theorems are for contrastive learning) and (2) as my opinion is the only strong opinion for the evaluation, I modified my evaluation on the empirical evaluation as "okay". As I am not certain about this part, I modified my confidence from 5 to 4. Certainly, more comparisons with other SSL methods will make this evaluation more stronger.
>
> Based on these criteria, I modified my score to borderline reject, which means I think this paper still has a large room for improvement, but at the same time, I will respect all the decisions by the other reviewers and area chair.

---

> > ### Author Response · Authors · 2023-11-23
> > **Thank you for raising your score**
> >
> > Thank you for your response. We are grateful that you have raised your score from 1 to 5.
> >
> > However, we noticed that you still have a few minor concerns about our work, and we will reply to them one by one:
> >
> > 1. We respect your opinion, but we have emphasized repeatedly that terms such as "encouraging" and "assuming the solutions" are commonly used in the literature, and we have already provided many references in our previous rebuttal.
> >
> > 2. Thank you for reading our rebuttal carefully and raising your evaluation of our paper.
> >
> > 3. Thank you for understanding and recognizing our standpoint.

---

### Official Review · Reviewer_cGgm · 2023-11-02

**Soundness:** 3 good
**Presentation:** 3 good
**Contribution:** 2 fair
**Rating:** 6
**Confidence:** 3

**Summary:**

The authors consider a theoretical framework to analyze and enhance self-supervised learning (SSL) methodologies utilizing matrix information theory. The work is particularly focused on providing a unified lens for examining both contrastive and feature decorrelation-based SSL paradigms through the application of matrix mutual information and joint entropy.

SSL, a significant branch of unsupervised learning, leverages unlabeled data to learn representations by predicting certain input parts from others. The authors' investigation into the utility of matrix mutual information in SSL is notable. By extending mutual information to the matrix domain, the manuscript aims to elucidate the dependencies among various features or representations within SSL models, shedding light on the information propagation mechanisms within neural networks. Additionally, the manuscript's exploration of joint entropy in the context of SSL is insightful. Assessing how the uncertainty in the input data influences the learning process and the quality of the learned representations can be crucial for enhancing model robustness and efficiency.

**Strengths:**

**Theoretical Innovation**: The manuscript presents a novel theoretical framework for analyzing self-supervised learning (SSL) methods through the prism of matrix information theory. The use of matrix mutual information and joint entropy is an innovative approach that could provide new insights into the dependencies between features and the propagation of information within neural networks. This theoretical advancement has the potential to deepen our understanding of SSL mechanisms, making it a significant contribution to the field.

**Unified Analysis for Diverse SSL Approaches**: By offering a unified analytical lens for both contrastive and feature decorrelation-based SSL paradigms, the paper bridges a gap in the current literature. This comprehensive approach allows for a more holistic understanding of SSL and its various implementations, enhancing the ability to compare and improve upon different methods within a common theoretical framework.

**Potential for Enhanced Robustness and Efficiency**: The exploration of joint entropy in SSL models addresses the critical aspect of input data uncertainty. By theoretically examining how this uncertainty affects the learning process, the paper lays the groundwork for developing more robust and efficient SSL algorithms that can better handle real-world data variability.

**Weaknesses:**

**Scalability of Information-Theoretic Measures**: A potential weakness could be the lack of a clear discussion on the scalability of the proposed matrix mutual information and joint entropy measures. Calculating these metrics can be computationally intensive, especially for large-scale datasets and high-dimensional feature spaces typical in self-supervised learning. Any insights on the computational overheads is much appreciated

**Questions:**

**Generalization to Diverse Architectures**: Your paper appears to focus on a specific class of self-supervised learning models. How generalizable is your matrix information-theoretic approach to other SSL architectures, such as transformer-based or recurrent neural networks? Can you provide empirical evidence or theoretical justification for the generalizability of your approach?

**Robustness and Sensitivity Analysis**: How robust are your matrix information-theoretic measures to variations in SSL hyperparameters, such as temperature in contrastive learning or weight decay? Could you provide a sensitivity analysis that examines the stability of your proposed metrics under different hyperparameter settings?

These questions are intended to probe the empirical validation of theoretical insights, the generalizability of the approach to various architectures, and the robustness of the proposed metrics to hyperparameter variations. Addressing these points could significantly strengthen the paper's contributions.

---

> ### Author Response · Authors · 2023-11-20
> **Response to reviewer cGgm**
>
> The three questions you raised are very insightful, and we will discuss them in details as follows.
>
> >Q1: Scalability of Information-Theoretic Measures: A potential weakness could be the lack of a clear discussion on the scalability of the proposed matrix mutual information and joint entropy measures. Calculating these metrics can be computationally intensive, especially for large-scale datasets and high-dimensional feature spaces typical in self-supervised learning. Any insights on the computational overheads is much appreciated
>
> A1: Thank you for your suggestion. **Though our paper mainly focus on give theoretical understanding of SSL using matrix information theory, we will discuss the computational aspect of it as follows.** Our metrics are calculated based on embeddings. In self-supervised learning, the dimension $d$ of embeddings usually does not exceed 1024, which is not that high dimensional. When we calculate the empirical covariance matrix, the size will always be $d \times d$, so the large scale dataset is not a bottleneck for calculating the information-theoretic measures. To reduce the computation burden, one may sample batches of samples from the data and approximately calculate the information-theoretic quantities. Thus the overall computation overhead of calculating matrix information-theoretic measures in SSL will be moderate.
>
> >Q2: Generalization to Diverse Architectures: Your paper appears to focus on a specific class of self-supervised learning models. How generalizable is your matrix information-theoretic approach to other SSL architectures, such as transformer-based or recurrent neural networks? Can you provide empirical evidence or theoretical justification for the generalizability of your approach?
>
> A2: Thank you for your feedback. Our improvement M-MAE is agnostic to the underlying structure and solely focuses on the loss function. **The reason we only implemented the transformer is that MAE and its follow-up works depend on the transformer architecture, not because our method is reliant on it.** Recurrent neural networks would be interesting to explore, but we haven't come across any papers that demonstrate how to incorporate them in vision SSL. If you can provide some references, we would be willing to experiment with them.
>
> >Q3: Robustness and Sensitivity Analysis: How robust are your matrix information-theoretic measures to variations in SSL hyperparameters, such as temperature in contrastive learning or weight decay? Could you provide a sensitivity analysis that examines the stability of your proposed metrics under different hyperparameter settings?
>
> A3: That is a great question!  We conduct empirical studies on a famous contrastive learning method SimCLR which uses the InfoNCE loss. One of the important hyper-parameters in SimCLR is the temperature in InfoNCE loss. We plot the matrix mutual information and matrix joint entropy during the pretraining of SimCLR on CIFAR-10 with different temperatures. The matrix mutual information is in the Figure 1 (https://i.postimg.cc/J7bhbTdm/matrix-mutual-information.png). The matrix joint entropy is in Figure 2 (https://i.postimg.cc/Pr0yD5LP/matrix-joint-entropy.png). We set temperatures as 0.3, 0.5, 0.7. **From the Figure 1 and 2, we can observe that the increase of matrix mutual information or matrix joint entropy during training ties closely with the final accuracy.** As temperature = 0.3 outperforms 0.5 and 0.7 in KNN accuracy, it also has the biggest matrix mutual information and matrix joint entropy value.

---

> ### Author Response · Authors · 2023-11-20
> **We would be grateful if you could take a look at the response**
>
> Dear Reviewer cGgm:
>
> We sincerely appreciate your valuable time devoted to reviewing our manuscript. We would like to gently remind you of the approaching deadline for the discussion phase. We have diligently addressed the issues you raised in your feedback, providing detailed explanations. For instance, we have added experiment on example the robustess of matrix information theoretic quantites on hyper-parameters as required. We use 3 different temperatures in the SimCLR method and plot the matrix mutual information and matrix joint entropy during the run. Would you kindly take a moment to look at it?
>
> We are very enthusiastic about engaging in more in-depth discussions with you.

---

### Official Review · Reviewer_znaA · 2023-11-05

**Soundness:** 3 good
**Presentation:** 2 fair
**Contribution:** 3 good
**Rating:** 8
**Confidence:** 3

**Summary:**

The paper discusses the information flow of three mainstream self-supervised learning methods: contrastive learning; feature de-correlation; and masked auto-encoding. It successfully connects all three methods with matrix information theory and thus offering a unified view. And beyond that it proposes to add an additional term to the original MAE loss. The loss regularizes the latent codes and is shown to be helpful on image classification tasks.

**Strengths:**

+ The paper introduces matrix information theory to understand and connect mainstream methods in self-supervised learning, which is a very meaningful and valuable contribution.
+ The writing is fairly clear, although I did not delve into the mathematical details, I believe they are sounds.
+ The initial results on image classification (both the linear-proving and the fine-tuning results) are great.

**Weaknesses:**

- While the initial empirical results are great, I do hope to see the final results after having a complete run on MAE. The current version of the paper uses U-MAE's implementation and the hyper-parameters (e.g., batch size) do not follow the settings in MAE. This can cause some discrepancies. MAE's ViT-L, after convergence, can achieve an accuracy of ~85.5 on ImageNet. While the paper's result is promising, it is unclear the trend can still hold. So I would be curious to see. If it is too much of a computation burden, I am fine to see results on CIFAR-100.
- There are some definitions used before they are defined (e.g., TCR is defined in the appendix). It would be great to at least point to them.

**Questions:**

* The TCR loss on the latent codes, how does it contribute to the entropy of the model? Is there a similar plot one can show as the training of M-MAE proceeds to Figure 1/2?

---

> ### Author Response · Authors · 2023-11-19
> **Response to reviewer znaA**
>
> >Q1: While the initial empirical results are great, I do hope to see the final results after having a complete run on MAE. The current version of the paper uses U-MAE's implementation and the hyper-parameters (e.g., batch size) do not follow the settings in MAE. This can cause some discrepancies. MAE's ViT-L, after convergence, can achieve an accuracy of ~85.5 on ImageNet. While the paper's result is promising, it is unclear the trend can still hold. So I would be curious to see. If it is too much of a computation burden, I am fine to see results on CIFAR-100.
>
> A1: Thank you for your suggestion. We do experiments mainly based on U-MAE for two reasons. One is that it improves the accuracy of MAE based on a modification of MAE's loss function and our work also changes the loss function by adding an additional entropy regularizer. **The other is that, as we have shown in our initial manuscript, U-MAE can be seen as a second-order approximation of our M-MAE loss, making it a much-related baseline compared to MAE.** We did our initial experiments following the configuration of U-MAE, so we only implemented 200 epochs.
>
> We have conducted experiments on CIFAR-100. The hyper-parameters are similar to U-MAE, and we set $\mu=1$ and pretrain CIFAR-100 for 1000 epochs. M-MAE may use hyperparameters that are not identical to U-MAE to fully reflect its potential. However, due to time constraints, we were unable to extensively search for these hyperparameters. We believe that with more reasonable hyperparameters, M-MAE can achieve even better results. As shown in the table below, our method performs remarkably well, even without an exhaustive hyperparameter search.
>
> | Method | linear@1 | linear@5 | finetune@1 | finetune@5 |
> | --- | --- | --- | --- | --- |
> | M-MAE(vit-base) | **60.9** | **88.1** | 83.8  | **97.0**  |
> | U-MAE(vit-base) | 59.7 | 86.8 | 84.1 | 96.5 |
> | MAE(vit-base) | 59.2 | 86.8 | 84.5 | 96.6 |
>
> >Q2: There are some definitions used before they are defined (e.g., TCR is defined in the appendix). It would be great to at least point to them.
>
> A2: Thank you for your feedback. **The TCR is already defined in definition 3.4 of the initial manuscript, which is before they are used.**
>
> >Q3: The TCR loss on the latent codes, how does it contribute to the entropy of the model? Is there a similar plot one can show as the training of M-MAE proceeds to Figure 1/2?
>
> A3: Thank you for your question. In our paper, the entropy of the model means the matrix entropy of the latent codes' covariance matrix. As we show in the paper, matrix entropy is directly related with effective rank. This definition of model's entropy aligns with the literatures that use effective rank to measure the dimension of the model's representation [1] [2]. For the matrix entropy plot you required, we plot it along our pretraining on CIFAR-100 (Figure is shown in https://i.postimg.cc/tT48m774/matrix-entropy.png). **From the Figure, it is clear that M-MAE has an increasing entropy during training, which aligns well with our theory.**
>
>
>
>
> [1]: How Mask Matters: Towards Theoretical Understandings of Masked Autoencoders, Qi Zhang, Yifei Wang, Yisen Wang, NeurIPS 2022
>
> [2]: RankMe: Assessing the Downstream Performance of Pretrained Self-Supervised Representations by Their Rank, Quentin Garrido, Randall Balestriero, Laurent Najman, Yann LeCun, ICML 2023

---

> ### Author Response · Authors · 2023-11-20
> **We would be grateful if you could take a look at the response**
>
> Dear Reviewer znaA:
>
> We sincerely appreciate your valuable time devoted to reviewing our manuscript. We would like to gently remind you of the approaching deadline for the discussion phase. We have diligently addressed the issues you raised in your feedback, providing detailed explanations. For instance, we have added experiments on CIFAR-100 as required. We pretrain the model for 1000 epochs to better show the effectiveness of our M-MAE method in the longer run. Would you kindly take a moment to look at it?
>
> We are very enthusiastic about engaging in more in-depth discussions with you.

---

> > ### Comment · Reviewer_znaA · 2023-11-23
> >
> > Thanks, I have checked the response.
> >
> > I think the explanations are clear, so thank you!
> >
> > Regarding the experiments on CIFAR-100. I think they are not as solid as ImageNet ones but okay-ish (e.g., fine-tuning degradation is a *big* concern to me, as methods like MAE are highlighting their usefulness when it comes to fine-tuning, and not linear evaluation).
> >
> > However, I think the bigger issue is on the theoretical side. I admit that I haven't checked the proofs carefully, so please first address the concerns from other reviewers.

---

> > > ### Author Response · Authors · 2023-11-23
> > >
> > > Thank you for your detailed reading.
> > >
> > > ---
> > >
> > > **About the experiment part**:
> > >
> > > As we have discussed, our method performs well, even without an exhaustive hyperparameter search on CIFAR-100. We will tune better hyper-parameters and will update the result once we get an increase in top-1 fine-tuning accuracy.
> > >
> > > ---
> > >
> > > **About the theoretical part**:
> > >
> > > Reviewer j36B appears to have a **misunderstanding** regarding the common optimization terminology and the details of our proof. In addressing these concerns, we offered comprehensive clarifications on two aspects: 1) what "a minimum (value) of n" means in an optimization problem, and 2) a straightforward derivation based on the definition. **Regrettably, despite our thorough response, we have not received any subsequent feedback.**
> > >
> > >
> > >
> > >
> > > For Reviewer qQv9 (who gave us a score of 1), we've engaged in extensive, multi-round discussions with them. However, we believe that much of this discourse stems not from issues in our paper but from a misinterpretation of our work. After our explanations, **the reviewer acknowledged they have misunderstandings.** We are concerned that the lengthy response might mislead others. To save your time, we highlight the **most** critical instances where we believe their comments demonstrate **a severe bias**:
> > >
> > > 1. **Typo-based misunderstanding**: The initial review contains 6 issues, 4 of which **originated from a typo** (K1=Z1, K2=Z2), and the rest 2 are experimental issues. The reviewer initially tried to dismiss our work entirely, **but accepted our explanation and did not raise these issues further.**
> > >
> > > 2. **Misinterpretation of simple math concepts**: The reviewer misinterpreted the simple expression $(z^{(1)}_i )^T z^{(2)}_j \approx 0$ as $z^{(1)}_i \approx z^{(2)}_j \approx 0$. **This misunderstanding led to misinterpretation in four out of seven steps of our proof. Despite this, after we clarified, the reviewer acknowledged their error with a comment**, "Sorry for my typos at the beginning of the response (e.g., $z\approx 0$)." (See our A8 and the beginning of https://openreview.net/forum?id=WfjJOEfAf7&noteId=c62yc0i0YA)
> > >
> > > 3. **Selective Quoting**. **What we actually stated was**, "As we point out, in traditional information theory, when the Siamese structure degenerates into a single branch like MAE, the joint entropy and mutual information are exactly the entropy." The reviewer, however, **selectively quoted only part of this, challenging our statement without considering its full context. He later admitted his mistake after we pointed the issue of selective quoting out.** (See our A12 and the reviewer's response in https://openreview.net/forum?id=WfjJOEfAf7&noteId=c62yc0i0YA)
> > >
> > >
> > > 4. **Misrepresentation of Facts**: **The reviewer falsely claimed that we equated two loss functions, which we never stated.** (See our A9, A18)
> > >
> > >
> > > 5. From an experimental standpoint, the reviewer **failed to understand the paradigm of self-supervised learning.** They **persisted** in believing our method was overly sensitive to hyperparameters, **despite our references to significant works** like SimCLR, VICReg, and MAE. Moreover, **they** stated that "MAE is evaluated by fine-tuning rather than linear probing, but I **don't think it makes a different** principle from others.", which **we find debatable and unfairly dismissive** of MAE's innovative nature. (See our A13, A25, A6, A22, and their response to our A6 in https://openreview.net/forum?id=WfjJOEfAf7&noteId=oRPRl4hGPE)
> > >
> > >
> > > **In addition to these points, there were several other issues raised by the reviewer qQv9,** which we have highlighted in bold in our previous messages for your reference. We believe that much of the reviewer's critique stems from misunderstandings or misinterpretations of our work.
> > >
> > > ---
> > >
> > > We appreciate your attention to these matters and **are available for any further clarifications needed.**

---

### Author Response · Authors · 2023-11-18
**General response**

We would like to thank the reviewers and area chairs for their hard work in reviewing our paper. We have made a minor revision to our manuscript based on the reviewers' feedback. The changes are as follows:

1. We correct the misname of "Theorem" to "Proposition" in Appendix A.
2. We correct the typos. For example, $K_1 = Z_1$, $K_2 = Z_2$ in Appendix A's proof of Theorem 3.2 and 3.8 to $K_1 = Z^T_1Z_1$ and $K_2 = Z^T_2Z_2$. This typo is made when we are dealing with spectral contrastive loss, which has similar form with Barlow Twins loss. As we set $K_1$ and $K_2$ correctly in the proof of Barlow Twins, we make this typo in spectral contrastive loss. Now, we have corrected it.
3. We cite the survey paper "A Survey on Masked Autoencoder for Self-supervised Learning in Vision and Beyond".
4. We change the statement of theorems 3.2 and 3.8, For example, we change the statement of our theorems from "Barlow twins and spectral contrastive learning seek the maximal mutual information" into "The optimal point of Barlow twins and spectral contrastive learning losses maximize the matrix mutual information". The proof basically unchanged except all the $\approx$ are changed into $=$, from the property of optimal point.
5. We add a remark at the end of Appendix A showing that the results can be generalized to sample contrastive and dimension contrastive methods defined in [1].

[1]: On the duality between contrastive and non-contrastive self-supervised learning, Quentin Garrido, Yubei Chen, Adrien Bardes, Laurent Najman, Yann LeCun, ICLR 2023 (notable-top-5%, ICLR 2023 Outstanding Paper Honorable Mention)

---

### Meta-Review · Area_Chair_yhYh · 2023-12-10

**Metareview:**

This paper uses matrix information theory to formulate contrastive and non-contrastive self-supervised learning methods. This is an interesting mathematical insight and the paper does a wonderful job of describing the insight in detail. The paper also proposes a matrix variational auto-encoder loss that minimizes the reconstruction loss in addition to the coding rate. This paper resulted in an extensive discourse between the reviewers and the authors. The authors are advised to consider these comments carefully as they revise the manuscript.

**Justification For Why Not Higher Score:**

Although the discussion resulted in a number of changes to the paper which have improved it, e.g, additional experiments and clarifications of the theory, the intellectual contributions of the paper still seem unclear. Section 3 discusses a unified formulation for contrastive and non-contrastive methods using matrix entropies. Section 4 is quite unrelated to this because it uses a matrix entropy-based term to fit a masked auto-encoder. It would be advisable to focus on a single problem and study it in depth, in one paper.

**Justification For Why Not Lower Score:**

N/A

---

### Decision · Program_Chairs · 2024-01-16

Reject